# Ral GTPases in Schwann cells promote radial axonal sorting in the peripheral nervous system

Andrea Ommer[1] ⓘ, Gianluca Figlia[1] ⓘ, Jorge A. Pereira[1], Anna Lena Datwyler[1], Joanne Gerber[1], Jonathan DeGeer[1], Giovanna Lalli[2], and Ueli Suter[1] ⓘ

**Small GTPases of the Rho and Ras families are important regulators of Schwann cell biology. The Ras-like GTPases RalA and RalB act downstream of Ras in malignant peripheral nerve sheath tumors. However, the physiological role of Ral proteins in Schwann cell development is unknown. Using transgenic mice with ablation of one or both Ral genes, we report that Ral GTPases are crucial for axonal radial sorting. While lack of only one Ral GTPase was dispensable for early peripheral nerve development, ablation of both RalA and RalB resulted in persistent radial sorting defects, associated with hallmarks of deficits in Schwann cell process formation and maintenance. In agreement, ex vivo–cultured Ral-deficient Schwann cells were impaired in process extension and the formation of lamellipodia. Our data indicate further that RalA contributes to Schwann cell process extensions through the exocyst complex, a known effector of Ral GTPases, consistent with an exocyst-mediated function of Ral GTPases in Schwann cells.**

## Introduction

In peripheral nerves, myelin produced by Schwann cells (SCs) enhances conduction velocity by enabling saltatory conduction of action potentials. SCs are derived from neural crest cells and undergo a series of differentiation steps before forming myelin (Jessen et al., 2015). During the process known as radial sorting, SCs proliferate and expand cellular extensions into bundles of unsorted axons to detach individual axons and establish the one-to-one relationship required for myelination (Webster et al., 1973). Axons with a diameter of <1 µm remain in bundles, and SCs in contact with these axons differentiate into non-myelinating SCs (Griffin and Thompson, 2008; Feltri et al., 2016).

Radial sorting of axons and myelination are tightly regulated and depend on signals from axons as well as the extracellular matrix (Ghidinelli et al., 2017). Prior to radial sorting, immature SCs deposit a basal lamina, which binds to laminin receptors on SCs, including integrin β1, dystroglycan, and GPR126. Loss of any of these laminin-binding proteins leads to defects in radial sorting (Feltri et al., 2002, 2016; Berti et al., 2011; Petersen et al., 2015). For loss of integrins, these defects have been largely attributed to SCs failing to extend the cellular protrusions that are necessary for axonal sorting. The associated signaling is mainly mediated by small Rho GTPases, with a prominent role of Rac1 (Feltri et al., 2002; Benninger et al., 2007; Nodari et al., 2007; Pereira et al., 2009). Small GTPases are signaling proteins that cycle between an active GTP-bound and an inactive GDP-bound state. Their activity levels are tightly controlled by guanine nucleotide-exchange factors (GEFs) and GTPase-activating proteins (GAPs; Wennerberg et al., 2005). Besides the Rho GTPase family, Ras is another small GTPase with a prominent function in diseased SCs. Hyperactive Ras due to mutations of the Ras-GAP neurofibromin 1 is a major cause of malignant transformation of SCs, with active Ras exerting some of its oncogenic potential through activation of the Ras-like GTPase RalA (Bodempudi et al., 2009).

The Ral family of small GTPases consists of the two members RalA and RalB (Chardin and Tavitian, 1986, 1989). Their activity is regulated by Ral GEFs and GAPs, which act mainly downstream of Ras and Akt (Gentry et al., 2014). There are four major known effectors of Ral GTPases: Ral-binding protein 1 (RalBP1; Cantor et al., 1995), phospholipase D1 (PLD1; Jiang et al., 1995), and the exocyst complex components Exoc2 and Exoc8 (Brymora et al., 2001). Through interaction with these effectors, Ral proteins have been implicated in a wide range of cellular functions and signaling cascades known to be critical for SC development, such as proliferation (Kashatus et al., 2011), receptor-mediated endocytosis (Jullien-Flores et al., 2000), mTOR signaling (Xu et al., 2011), and vesicle targeting (Teodoro et al., 2013).

In peripheral nerves, a study aimed at identifying novel myelination-related molecules revealed that RalA mRNA levels peak at postnatal day 15 (P15; Patzig et al., 2011). In addition, RalA protein was found in both cytoplasm and pseudopods of cultured SCs, while RalB was only present in pseudopods

[1]Institute of Molecular Health Sciences, Department of Biology, ETH Zurich, Zurich, Switzerland;   [2]Wolfson Centre for Age-Related Diseases, King's College London, London, UK.

Correspondence to Ueli Suter: usuter@cell.biol.ethz.ch;   G. Lalli's present address is UK Dementia Research Institute, University College London, London, UK.

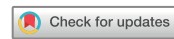

(Poitelon et al., 2015). Of further interest in the context of Ral effectors, the exocyst complex subunit Exoc4 has been shown to be involved in membrane addition during myelin formation, in combination with Myotubularin-related protein 2 (Mtmr2) and Discs large homolog 1 (Bolis et al., 2009).

The prominent involvement of Ral GTPases in several signaling pathways that are recognized to be crucial for SCs implies that these proteins can potentially act as an important signaling hub in SC biology. Indeed, using appropriate transgenic mice, we found that RalA and RalB are functionally redundant in the development of SCs, while simultaneous elimination of both proteins caused severely arrested axonal sorting. Our data indicate that this defect is caused by deficits that affect SC processes. In this context, we provide evidence that the exocyst complex is a component of Ral-mediated signaling in the regulation of SC process extensions. Taken together, our results identify and establish Ral GTPases as crucial regulators of SC development.

## Results

### SC-specific deletion of RalA in RalB$^{-/-}$ animals causes mild motor impairments in transgenic mice

To study the functions of RalA and RalB in developing peripheral nerves, mice deficient for one or both of these GTPases were generated. Since constitutive deletion of RalA is embryonically lethal (Peschard et al., 2012), we ablated RalA specifically in SCs. To achieve this goal, we crossed mice containing loxP sites flanking exons 2 and 3 of the *Rala* gene (Peschard et al., 2012) with transgenic mice expressing Cre recombinase under control of the SC-specific *Mpz* promoter (Feltri et al., 1999) to generate $Mpz^{Cre/+}$:$Rala^{fl/fl}$ mice (Fig. 1 A, left). Constitutive $Ralb^{-/-}$ mice (Fig. 1 A, right) have been described previously with no detectable abnormalities (Peschard et al., 2012). Using these alleles, double-mutant $Mpz^{Cre/+}$:$Rala^{fl/fl}$:$Ralb^{-/-}$ mice were also produced. To check for efficiency of Ral GTPase depletion, we first analyzed lysates of P5 sciatic nerves by quantitative RT-PCR. A marked reduction of RalA mRNA levels was evident in $Mpz^{Cre/+}$:$Rala^{fl/fl}$ and $Mpz^{Cre/+}$:$Rala^{fl/fl}$:$Ralb^{-/-}$ compared with control and $Rala^{fl/fl}$:$Ralb^{-/-}$ mice (Fig. 1 B). As expected, RalB mRNA was not detectable in mice containing two constitutive $Ralb^{-}$ alleles. RalB levels were mildly increased in $Mpz^{Cre/+}$:$Rala^{fl/fl}$ mice compared with controls, with rather high variability (Fig. 1 C). We confirmed these observations on the protein level by Western blot analysis of P5 sciatic nerve lysates (Fig. 1, D–F). In contrast to the mRNA analyses, however, the amount of RalB protein was not increased in $Mpz^{Cre/+}$:$Rala^{fl/fl}$ mice (Fig. 1 F).

Single- ($Mpz^{Cre/+}$:$Rala^{fl/fl}$ and $Rala^{fl/fl}$:$Ralb^{-/-}$) and double-mutant ($Mpz^{Cre/+}$:$Rala^{fl/fl}$:$Ralb^{-/-}$) mice appeared healthy and indistinguishable from controls up to the age of 2 mo as assessed by visual inspection. CatWalk analysis revealed that double mutant mice had mildly impaired motor function (Fig. 1 G), which was apparent by a shortened distance between hindpaws (base of support, Fig. 1 H), an increased percentage of the step cycle that was spent on more than two paws (support, Fig. 1 I), and a shortened stride length (Fig. 1 J). Single-mutant

mice showed no significant differences compared with controls in this test.

### Loss of RalA in SCs in RalB$^{-/-}$ animals impairs radial sorting of axons, while loss of a single Ral GTPase is dispensable for early peripheral nerve development

Upon dissection, sciatic nerves of 2-mo-old double-mutant ($Mpz^{Cre/+}$:$Rala^{fl/fl}$:$Ralb^{-/-}$) mice appeared thinner and more translucent than those of single mutants ($Mpz^{Cre/+}$:$Rala^{fl/fl}$, $Rala^{fl/fl}$:$Ralb^{-/-}$) and controls (Fig. S1). To investigate if Ral-deficient mice display further morphological impairments in peripheral nerve development, we analyzed cross sections of sciatic nerves at different time points by EM (Fig. 2 A). We compared single mutants, double mutants, and controls at P5 and 2 mo of age. At P5, double-mutant, but not single-mutant, mice showed fewer myelinated axons (Fig. 2 B) and fewer sorted axons (i.e., myelinated plus not-myelinated axons at the 1:1 stage; Fig. 2 C). However, the number of not-myelinated sorted axons alone was not significantly changed (Fig. 2 D). As a prominent characteristic in double mutants, we found that bundles of unsorted axons occupied a larger part of the endoneurium compared with single mutants and controls (Fig. 2 E). These results indicate that SC-specific deletion of RalA paired with constitutive lack of RalB causes defects in the early stages of radial sorting. The typical features of anomalous radial sorting persisted, although as a less pronounced trait, in sciatic nerves of 2-mo-old double-mutant mice (Fig. 2, F–I). At this time point, we also found increased numbers of not-myelinated sorted axons with a diameter >1 µm in double-mutant nerves, a very rare feature in controls and single mutants (Fig. 2 H). One interpretation of these data is that aberrantly late events of axon sorting occur in double mutants, with possible contributions by demyelination already in young adults. Taken together, our findings show that radial sorting is abnormally delayed in double-mutant mice, while this process progresses normally in single mutants.

### Long-term lack of RalA in SCs in RalB$^{-/-}$ animals leads to complex peripheral nerve defects

Since the radial sorting defects improved between the age of P5 and 2 mo in $Mpz^{Cre/+}$:$Rala^{fl/fl}$:$Ralb^{-/-}$ mice (Fig. 2), we wondered if radial sorting would be completed in such double mutants later. Furthermore, we wanted to examine the long-term consequences of loss of Ral proteins in our mutants. Thus, we performed a comparative morphological analysis of sciatic nerves of 1-yr-old controls, single mutants ($Mpz^{Cre/+}$:$Rala^{fl/fl}$, $Rala^{fl/fl}$:$Ralb^{-/-}$), and double mutants ($Mpz^{Cre/+}$:$Rala^{fl/fl}$:$Ralb^{-/-}$). This evaluation revealed that radial sorting is not fully accomplished even in 1-yr-old double-mutant mice (Fig. 3 A). Occasional bundles of unsorted axons were still present at this age, together with a persistent reduction in the numbers of both myelinated axons and of all sorted axons (Fig. 3, B and C). Interestingly, the numbers of both myelinated (Fig. 3 B) and of all sorted axons (Fig. 3 C) decreased in double mutants from 2 mo to 1 yr of age, consistent with axonal loss. Furthermore, we observed onion bulb–like structures usually associated with thinly myelinated axons in 1-yr-old double-mutant mice (Fig. S2 A),

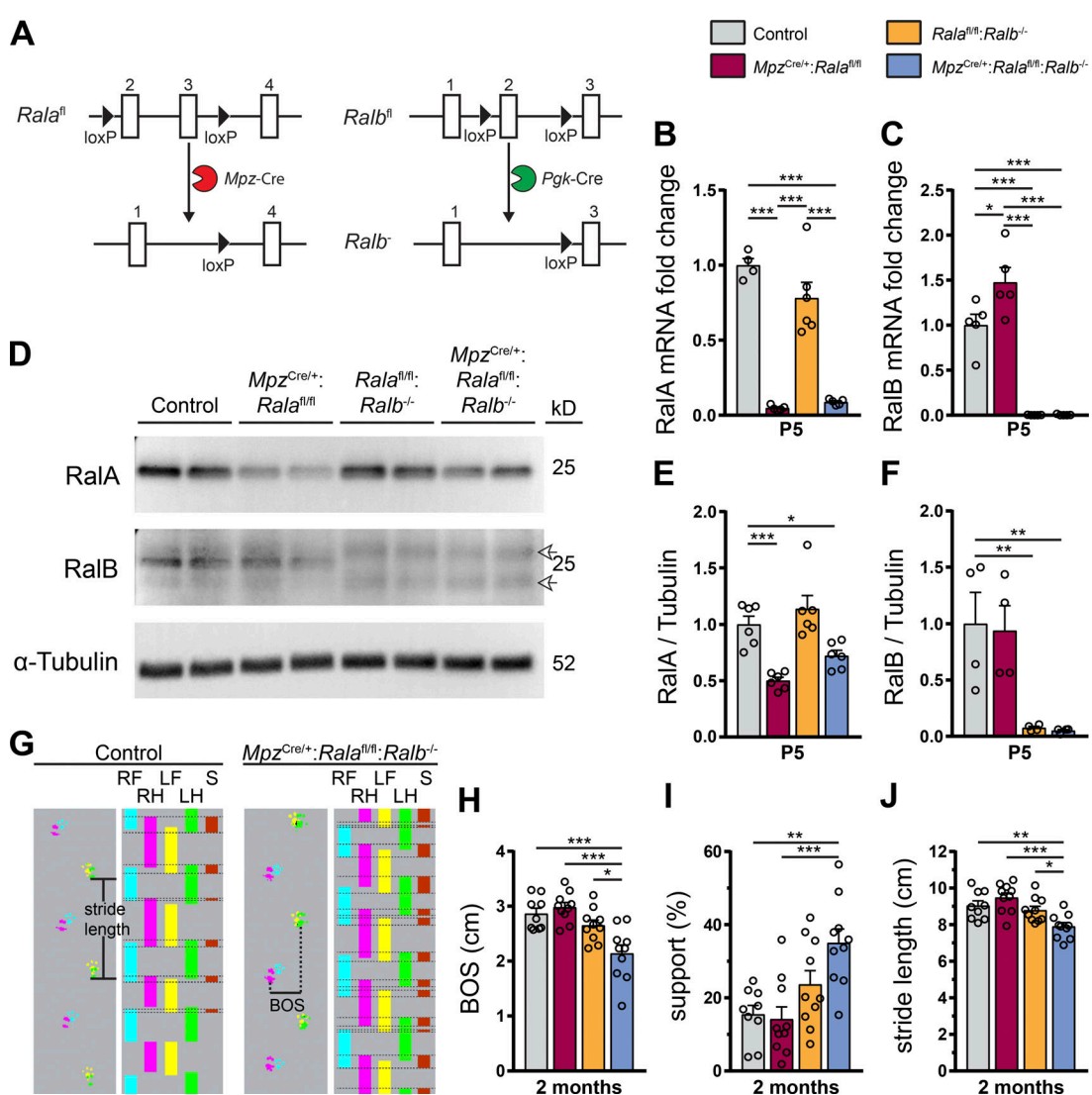

Figure 1. **Loss of RalA in SCs in RalB$^{-/-}$ animals leads to mildly impaired motor function. (A)** Schematic depicting floxed conditional alleles for RalA (*Rala*$^{fl}$) and RalB (*Ralb*$^{fl}$) as well as the constitutive knockout allele for RalB (*Ralb*$^{-}$) obtained by Cre-mediated recombination. **(B and C)** mRNA expression of RalA (B) and RalB (C) in P5 sciatic nerves, analyzed by quantitative RT-PCR, and shown as fold change relative to controls after normalization to β-actin. *n* = 4 (control in B), 6 (*Rala*$^{fl/fl}$:*Ralb*$^{-/-}$ in B), or 5 (all other groups) mice per genotype. One-way ANOVA with Tukey's multiple comparisons test. **(D–F)** Western blot of RalA and RalB protein expression in P5 sciatic nerve lysates. Arrows in D indicate unspecific antibody binding. Quantification of protein expression relative to tubulin is shown for RalA (E) and RalB (F). *n* = 6 (E) or 4 (F) mice per genotype. One-way ANOVA with Dunnett's multiple comparisons test. Note the limited specificity of the RalB antibody used in this analysis. **(G)** Exemplary traces of footprints and footfall patterns of 2-mo-old control and *Mpz*$^{Cre/+}$:*Rala*$^{fl/fl}$:*Ralb*$^{-/-}$ mice obtained by CatWalk analysis. BOS, base of support; LF, left front; LH, left hind; RF, right front; RH, right hind; S, support (quantified in I). **(H–J)** Quantification of CatWalk analysis of 2-mo-old mice. Shown are base of support (BOS; distance between hindpaws; H), support (percentage of the step cycle spent on more than two paws; I), and stride length (J). *n* = 9 (only control) or 10 mice per genotype. One-way ANOVA with Tukey's multiple comparisons test. All data are shown as mean ± SEM. *, P < 0.05; **, P < 0.01; ***, P < 0.001.

features that are commonly interpreted as signs of demyelination and incomplete remyelination (Dyck and Thomas, 2005). Consistent with the occurrence of demyelination, the number of not-myelinated sorted axons with calibers >1 µm increased from 2-mo-old to 1-yr-old double mutants (Fig. S2 B), although late axonal sorting may also contribute to this observation. Furthermore, sciatic nerves of double-mutant mice showed hypomyelination of large caliber axons at both 2 mo and 1 yr of age, while hypermyelinated small-caliber axons were evident especially at the age of 2 mo (Fig. 3, D and E). We also observed slightly more myelin abnormalities (i.e., infoldings, outfoldings,

tomacula, and detached myelin sheaths) in double mutants compared with single mutants and controls at both time points (Fig. 3 A, arrows; and Fig. S2, C and D). Regarding single mutants, no significant morphological changes were present in 1-yr-old *Mpz*$^{Cre/+}$:*Rala*$^{fl/fl}$ mice. However, 1-yr-old single mutant *Rala*$^{fl/fl}$:*Ralb*$^{-/-}$ mice displayed mild hypermyelination across all axonal sizes (Figs. 3 E and S3 A). As these mice lack RalB expression in all cell types, including neurons, hypermyelination might be due to age-associated shrinking of axons, since no hypermyelination was present in 2-mo-old *Rala*$^{fl/fl}$:*Ralb*$^{-/-}$ mice (Fig. 3 D). However, we did not notice a prominent shift in the

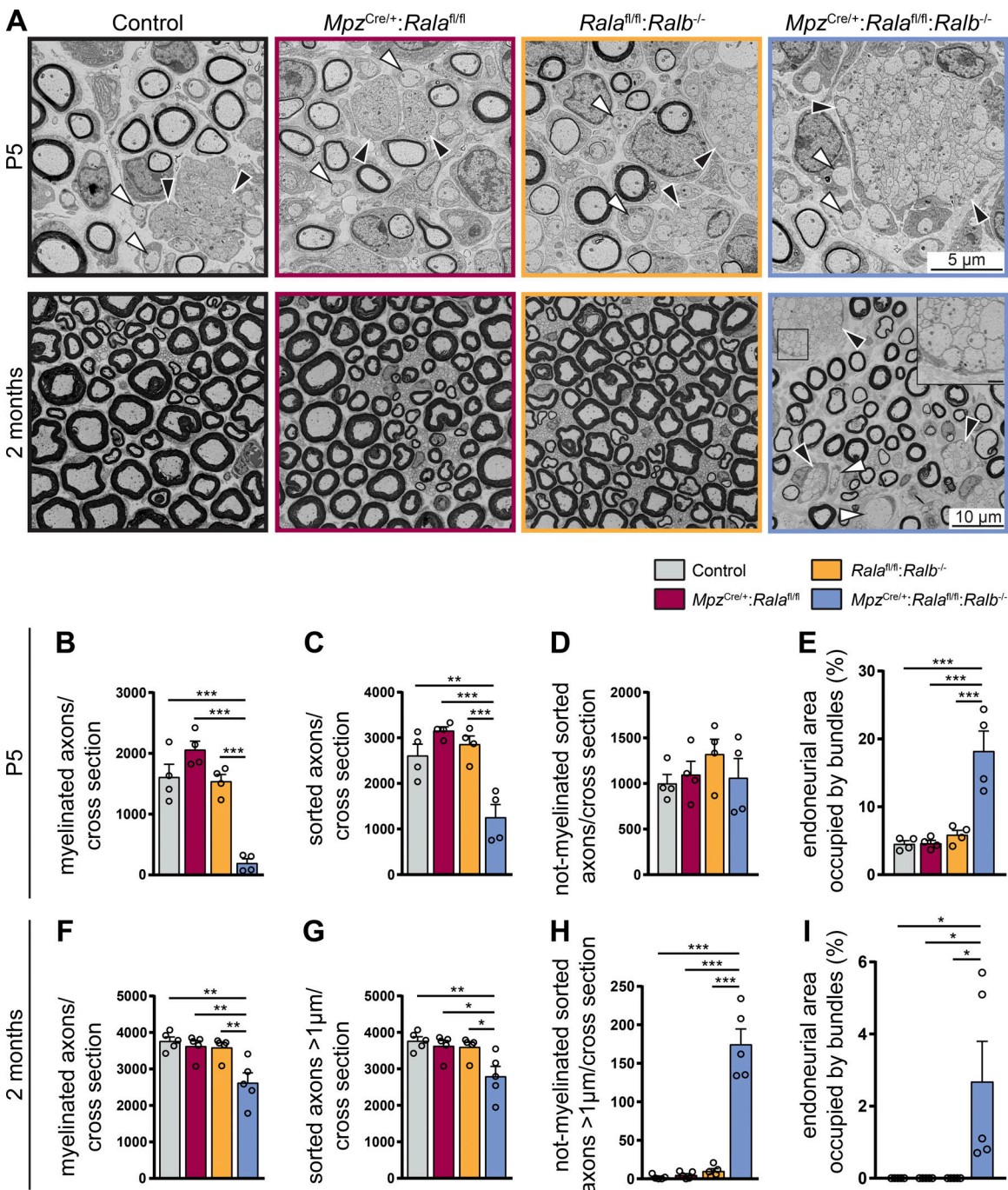

Figure 2. **Absence of RalA in SCs specifically and RalB ubiquitously impairs radial sorting. (A)** Electron micrographs of P5 sciatic nerves (top) and 2-mo-old (bottom) control, *Mpz*^Cre/+^:*Rala*^fl/fl^, *Rala*^fl/fl^:*Ralb*^−/−^, and *Mpz*^Cre/+^:*Rala*^fl/fl^:*Ralb*^−/−^ mice. Bundles of unsorted axons are indicated by black arrowheads. White arrowheads point to sorted but not-myelinated fibers. Scale bar of inset: 1 µm. **(B–E)** Quantification of morphological features of P5 sciatic nerves. The number of myelinated axons (B), all sorted axons (C), and not-myelinated but sorted axons (D) was determined per cross section. The area of endoneurium occupied by bundles of unsorted axons was quantified relative to the total endoneurial area (E). *n* = 4 mice per genotype; one complete nerve cross section per animal was analyzed. One-way ANOVA with Tukey's multiple comparisons test. **(F–I)** Quantification of morphological features of 2-mo-old sciatic nerves. The number of myelinated axons (F; same dataset is shown again in Fig. 3 B), all sorted axons with a diameter >1 µm (G; same dataset is shown again in Fig. 3 C), and not-myelinated but sorted axons with a diameter >1 µm (H; same dataset is shown again in Fig. S2 B) was determined per cross section. The area of the endoneurium occupied by bundles of unsorted axons was quantified relative to the total endoneurial area (I). *n* = 5 mice per genotype; one complete cross section per animal was analyzed. One-way ANOVA with Tukey's multiple comparisons test. All data are shown as mean ± SEM. *, P < 0.05; **, P < 0.01; ***, P < 0.001.

axonal diameter in the sciatic nerves of such mice by analyzing axon size distributions (Fig. S3 B). Overall, these data indicate that long-term lack of Ral proteins in SCs leads to complex

defects in peripheral nerves, with some resemblance to mouse models of peripheral neuropathies (Bolino et al., 2004; Bonneick et al., 2005; Horn et al., 2012).

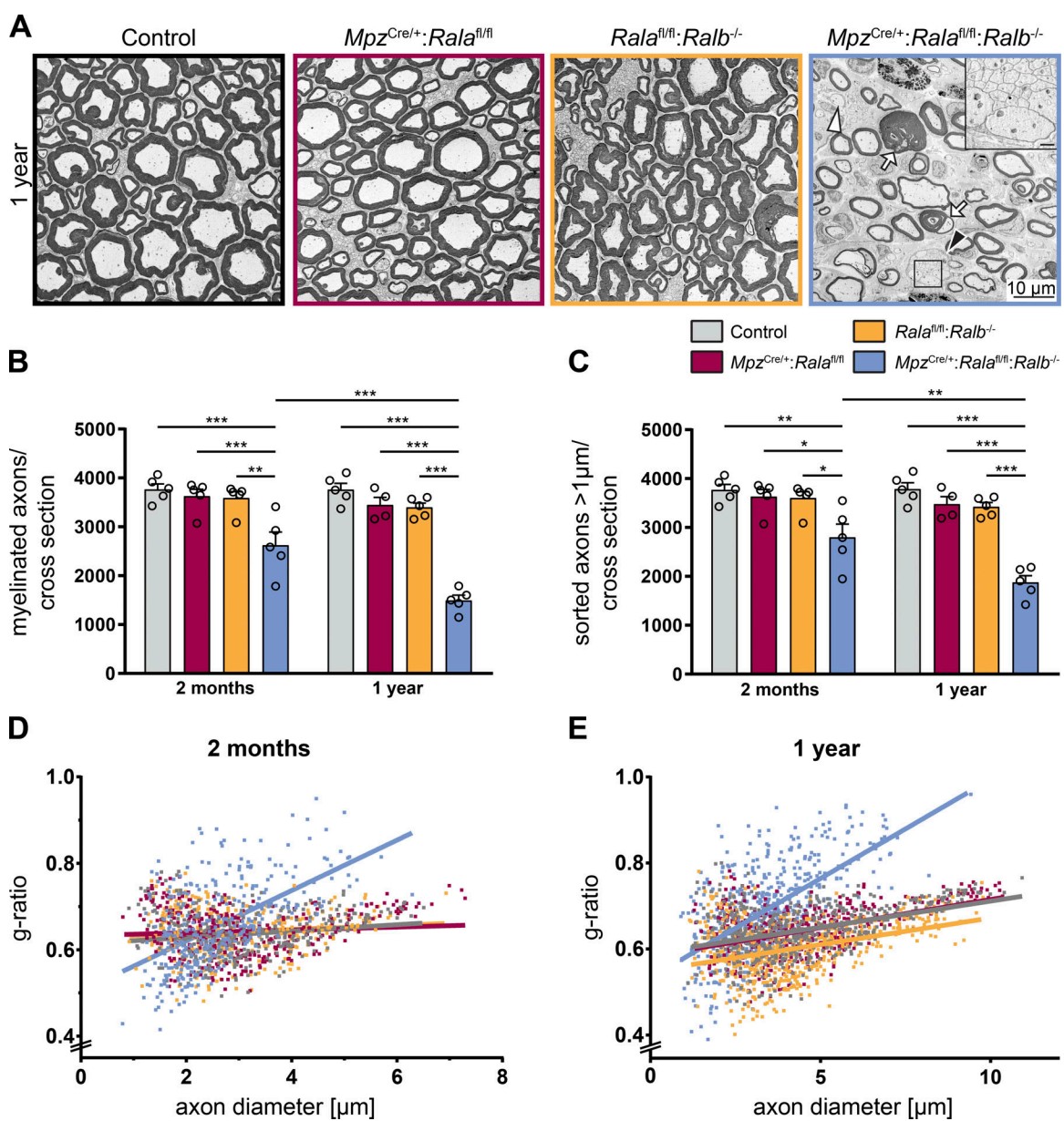

Figure 3. **Long-term loss of RalA in SCs and constitutive lack of RalB leads to complex peripheral nerve defects. (A)** Electron micrographs of sciatic nerves of 1-yr-old control, $Mpz^{Cre/+}:Rala^{fl/fl}$, $Rala^{fl/fl}:Ralb^{-/-}$, and $Mpz^{Cre/+}:Rala^{fl/fl}:Ralb^{-/-}$ mice. A bundle containing large-caliber, unsorted axons is indicated by a black arrowhead. A white arrowhead points to a sorted but not-myelinated axon with a diameter >1 µm. Arrows indicate examples of abnormal myelin profiles. Scale bar of inset: 1 µm. **(B and C)** Quantification of myelinated axons (B) and all sorted axons with a diameter >1 µm (C) per sciatic nerve cross section in 2-mo-old and 1-yr-old animals (dataset of 2-mo-old animals is shown in Fig. 2, F and G). $n$ = 4 (only $Mpz^{Cre/+}:Rala^{fl/fl}$ at 1 yr) or 5 mice per genotype, with one complete cross section per animal analyzed. Two-way ANOVA with Tukey's multiple comparisons test. All data are shown as mean ± SEM. *, P < 0.05; **, P < 0.01; ***, P < 0.001. **(D and E)** Distribution of g-ratio versus axon diameter in 2-mo-old (D) and 1-yr-old animals (E; same dataset is shown again in Fig. S3 A). Each dot represents an individual axon of $n$ = 4 (only $Mpz^{Cre/+}:Rala^{fl/fl}$ at 1 yr) or 5 mice per genotype, with ≥100 axons per animal analyzed from four randomly selected fields. Lines show linear regression calculated with GraphPad Prism (version 7.03).

## SC-specific lack of RalA and RalB is sufficient to impair radial sorting of axons

Since Ral GTPases are ubiquitously expressed and have relevant functions in different cell types (Gentry et al., 2014), we were concerned that constitutive deletion of RalB may affect our interpretations of the observed phenotype in developing $Mpz^{Cre/+}$: $Rala^{fl/fl}:Ralb^{-/-}$ mice with regard to the causative cell type. To verify that our previous findings on defective radial sorting

resulted from the absence of both Ral proteins exclusively and specifically in SCs, we used mice harboring a conditional $Ralb^{fl}$ allele to generate $Mpz^{Cre/+}:Ralb^{fl/fl}$ and $Mpz^{Cre/+}:Rala^{fl/fl}:Ralb^{fl/fl}$ animals. Focusing on P5 and 2-mo-old mice and using the same analytical design as before, we confirmed that radial sorting was also strongly impaired in $Mpz^{Cre/+}:Rala^{fl/fl}:Ralb^{fl/fl}$ mice by obtaining comparable data to $Mpz^{Cre/+}:Rala^{fl/fl}:Ralb^{-/-}$ mice (Figs. 2 and 4). No morphological alterations were

detected in SC-specific single mutants. We noted some differences in absolute numbers, but not in the relevant comparisons between the genotypes in the different mutant settings analyzed (Fig. 2 vs. Fig. 4). We consider it likely that these differences are due to variations in genetic backgrounds of the examined mice. Taken together, our results demonstrate that Ral proteins in SCs are essential to foster proper radial sorting of axons in peripheral nerves during early development.

## Ral double-mutant SCs display increased proliferation and normal rates of apoptosis

Ral GTPases have been implicated in cell cycle regulation as well as in the proliferation and transformation of tumor cells (Rossé et al., 2003; Bodempudi et al., 2009; Tazat et al., 2013). Before and during radial sorting, SCs have to proliferate to match axon and SC numbers. Alterations in this essential SC proliferation can cause defects in radial sorting (Feltri et al., 2016). Thus, we examined SC proliferation by injecting P5 single mutants ($Mpz^{Cre/+}$:$Rala^{fl/fl}$ and $Rala^{fl/fl}$:$Ralb^{-/-}$), double mutants ($Mpz^{Cre/+}$: $Rala^{fl/fl}$:$Ralb^{-/-}$), and controls with 5-ethynyl-2′-deoxyuridine (EdU) and analyzed the sciatic nerves 1 h later by colabeling for EdU and the SC marker Sox10, together with DAPI labeling (Fig. 5 A). All three mutants showed slightly higher cell numbers in sciatic nerves compared with controls, albeit with low significance (Fig. 5 B). Considering SCs specifically, the results were similar reaching a low level of significance for $Rala^{fl/fl}$: $Ralb^{-/-}$ mice only (Fig. 5 C). We have no definitive explanation for these findings but favor the interpretation of biological and/or technical variability without major biological relevance with regard to the function of Ral proteins. However, when we analyzed the incorporation of EdU in SCs, we found a robust increase in the fraction of EdU-positive SCs in double mutants compared with controls and single-mutant mice, with no differences detected between the latter and controls (Fig. 5 D). To complete the analysis, we also measured SC apoptosis on sciatic nerve cross sections stained for Sox10 and cleaved caspase-3 (CC3; Fig. 5 E), but we detected no differences in the fraction of apoptotic SCs among single mutants, double mutants, and controls (Fig. 5 F). Taken together, these data indicate that the observed substantial defects in radial sorting are most likely not due to a shortage in available SCs.

## Ral double-mutant SCs show hallmarks of process extension deficits in vivo and in vitro

Radial sorting defects in genetic mouse mutants have been attributed to deficiencies in SC process extension and stability. In this context, small GTPases of the Rho family have been established as key regulators (Benninger et al., 2007; Nodari et al., 2007; Pereira et al., 2009; Montani et al., 2014). Since Ral GTPases can influence the activity of Rac1 and Cdc42 (Cantor et al., 1995; Jullien-Flores et al., 1995; Park and Weinberg, 1995; Lee et al., 2014; Zago et al., 2017), we hypothesized that Ral GTPases may contribute to the regulation of SC process extension. In support of this hypothesis, high-magnification EM images of sciatic nerves of 2-mo-old double mutant ($Mpz^{Cre/+}$:

$Rala^{fl/fl}$:$Ralb^{-/-}$) mice revealed that bundles of unsorted axons were often surrounded by aberrant loops of redundant basal lamina associated with collagen (Fig. 6 A, left). In addition, those bundles appeared to be incompletely surrounded by SC processes (Fig. 6 A, middle). Similarly, the majority of sorted but not-myelinated axons were incompletely surrounded by SCs in double mutants (Fig. 6, A [right] and B). Such morphological features have also been reported upon Rac1 ablation in SCs (Benninger et al., 2007; Nodari et al., 2007). Thus, we measured Rac1 activity in lysates of P5 sciatic nerves to test for potential correlations. However, our analysis revealed increased Rac1 activity in double mutants compared with controls (Fig. 6, C and D), while the total amounts of Rac1 protein expression were not significantly changed (Fig. 6, C and E). On the molecular level, one interpretation of these results is a predicted reduced activity of the Ral GTPase effector RalBP1 in mutant SCs, since RalBP1 possesses GAP activity for Rac1 (Cantor et al., 1995; Matsubara et al., 1997). Taken together, our data indicate that Ral double-mutant SCs struggle with the formation and maintenance of their processes within these mutant sciatic nerves.

Anomalous morphological features in peripheral nerves such as the ones described above have generally been correlated with defective process formation of SCs in culture (Benninger et al., 2007; Nodari et al., 2007). Thus, we asked whether we would find similar defects in cultured SCs derived from our mouse mutants. To answer this question, we isolated SCs from sciatic nerves of P5 control and double-mutant ($Mpz^{Cre/+}$:$Rala^{fl/fl}$: $Ralb^{-/-}$) mice and plated them on laminin. Processes and lamellipodia were examined by staining for α-tubulin and F-actin after 1 d in vitro (DIV1; Fig. 7 A). Analysis of double-mutant–derived SCs revealed reduced numbers of both radial and axial lamellipodia compared with SCs isolated from control sciatic nerves (Fig. 7, B and C). In addition, we found reduced overall process lengths at DIV5 (Fig. 7 D). Our data show that the combined loss of RalA and RalB in ex vivo–cultured SCs impairs the formation of lamellipodia and accurate process extensions, providing a plausible explanation for the observed morphological defects in Ral double-mutant nerves.

## RalA promotes process extension in SCs through the exocyst complex

The exchange of single–amino acid residues in Ral GTPases can specifically abolish the interaction with one of the Ral effectors RalBP1, PLD1, or the exocyst complex components Exoc2 and Exoc8, thus preventing their downstream effects (Lalli and Hall, 2005). These mutations, if introduced in a constitutively active RalA mutant backbone, allow insights into whether a specific effector is required for a given Ral-mediated process. Thus, we used lentivirus-mediated expression of such mutants in cultured double-mutant and control SCs to determine whether one of these effectors would be necessary for accurate SC process extensions. As expected, a lentivirus encoding constitutively active RalA (RalA72L) was able to rescue the deficit in process length of double-mutant SCs (Fig. 8 A). In contrast, expression of a dominant-negative

Figure 4. **Ral proteins in SCs are essential to foster proper radial sorting of axons in peripheral nerves during early development. (A)** Electron micrographs of sciatic nerves of P5 (top) and 2-mo-old (bottom) control, *Mpz*^Cre/+^:*Rala*^fl/fl^, *Mpz*^Cre/+^:*Ralb*^fl/fl^, and *Mpz*^Cre/+^:*Rala*^fl/fl^:*Ralb*^fl/fl^ mice. Bundles of unsorted axons are indicated by black arrowheads. White arrowheads point to sorted but not-myelinated fibers. Scale bar of inset: 1 µm. **(B–E)** Quantifications of morphological features of P5 sciatic nerves. The number of myelinated axons (B), all sorted axons (C), and not-myelinated but sorted axons (D) was determined per cross section. The area of endoneurium that is occupied by bundles of unsorted axons was quantified relative to the total endoneurial area (E). *n* = 5 mice per genotype, with one complete cross section per animal analyzed. One-way ANOVA with Tukey's multiple comparisons test. **(F–I)** Quantifications of morphological features of 2-mo-old sciatic nerves. The number of myelinated axons (F), all sorted axons with a diameter >1 µm (G), and not-myelinated but sorted axons with a diameter >1 µm (H) was determined per cross section. The area of endoneurium that is occupied by bundles of unsorted axons was quantified relative to the total endoneurial area (I). *n* = 5 mice per genotype, with one complete cross section per animal analyzed. One-way ANOVA with Tukey's multiple comparisons test. All data are shown as mean ± SEM. ***, P < 0.001.

RalA mutant (RalA28N) could not increase the lengths of double-mutant SC processes to control levels (Fig. 8 B), consistent with the interpretation that RalA needs to be active to promote process extension in SCs (Jiang et al., 1995; Feig, 1999; Moskalenko et al., 2002). However, lentiviral expression of RalA72L D49N, a constitutively active RalA

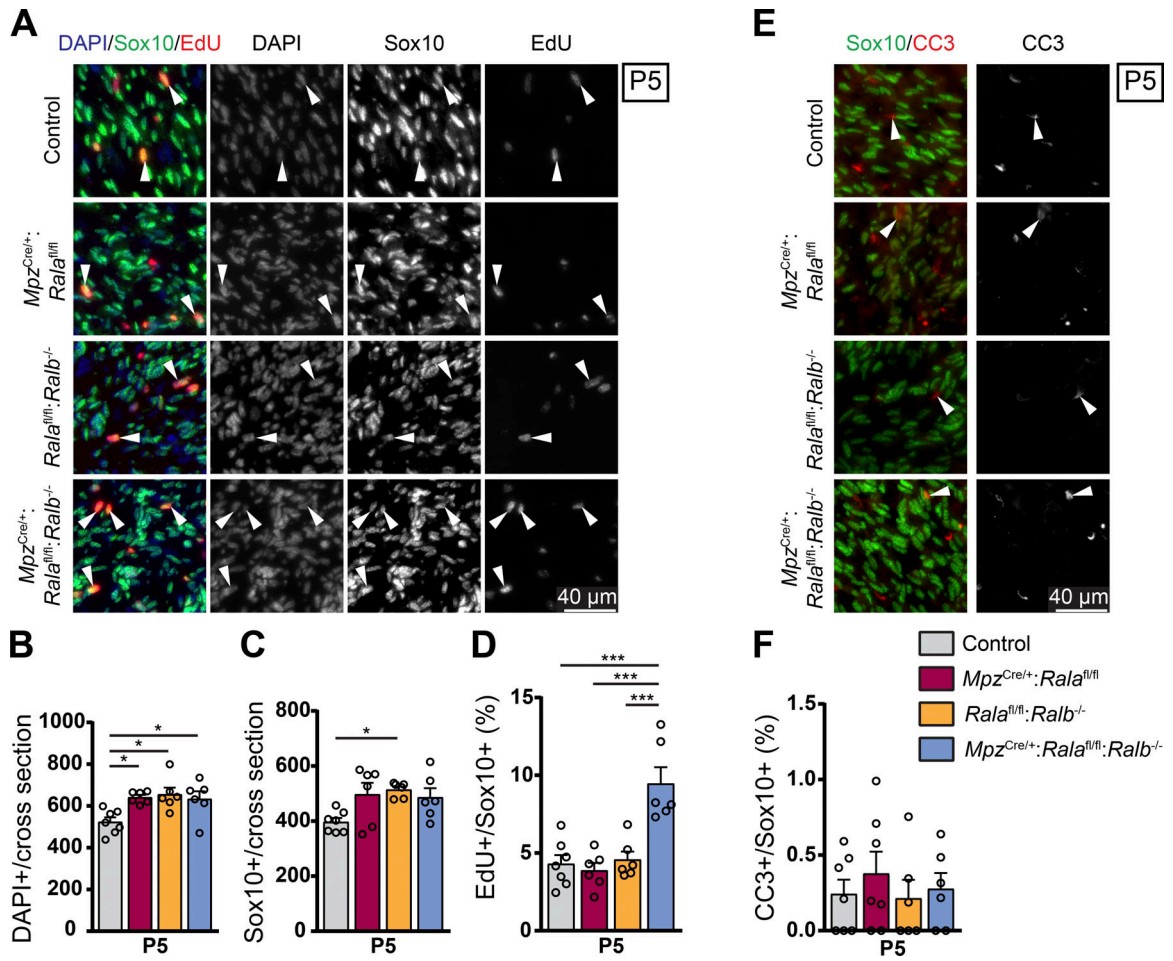

**Figure 5. SC proliferation is increased in *Mpz*<sup>Cre/+</sup>:*Rala*<sup>fl/fl</sup>:*Ralb*<sup>−/−</sup> sciatic nerves at P5. (A)** EdU detection on sciatic nerve cross sections of P5 control, *Mpz*<sup>Cre/+</sup>:*Rala*<sup>fl/fl</sup>, *Rala*<sup>fl/fl</sup>:*Ralb*<sup>−/−</sup>, and *Mpz*<sup>Cre/+</sup>:*Rala*<sup>fl/fl</sup>:*Ralb*<sup>−/−</sup> mice, injected with EdU 1 h before dissection. Colabeling with Sox10 by immunostaining and DAPI. Arrowheads highlight examples of EdU-positive SC nuclei. **(B and C)** Quantification of immunostainings of cross sections as depicted in A showing the number of nuclei (B) and SCs (C) per cross section. $n = 7$ (only control) or 6 mice per genotype, with at least one complete cross section analyzed per animal. One-way ANOVA with Tukey's multiple comparisons test. **(D)** Proliferating SCs (EdU<sup>+</sup> Sox10<sup>+</sup>) expressed as a fraction of all SCs (Sox10<sup>+</sup>). $n = 7$ (only control) or 6 mice per genotype, at least one complete cross section analyzed per animal. One-way ANOVA with Tukey's multiple comparisons test. **(E)** Immunostaining of CC3 and Sox10 on P5 sciatic nerve cross sections. Arrowheads highlight examples of CC3-positive SCs. **(F)** Quantification of immunostainings depicted in E showing the number of apoptotic SCs (CC3<sup>+</sup> Sox10<sup>+</sup>) as a fraction of all SCs (Sox10<sup>+</sup>). $n = 7$ (control, *Mpz*<sup>Cre/+</sup>:*Rala*<sup>fl/fl</sup>) or 6 (*Rala*<sup>fl/fl</sup>:*Ralb*<sup>−/−</sup>, *Mpz*<sup>Cre/+</sup>:*Rala*<sup>fl/fl</sup>:*Ralb*<sup>−/−</sup>) mice per genotype, at least one complete cross section analyzed per animal. One-way ANOVA with Tukey's multiple comparisons test. All data are shown as mean ± SEM. *, $P < 0.05$; ***, $P < 0.001$.

that is unable to interact with RalBP1, was able to rescue the process extension deficit of double-mutant SCs (Fig. 8 C). Thus, binding to the RalBP1 effector appears not to be majorly involved in RalA-mediated process extension. We also observed a rescue by expressing RalA72L ΔN11, a constitutively active RalA uncoupled from interaction with PLD1 (Fig. 8 D), ruling out a major contribution of this interaction. In contrast, expression of RalA72L D49E, which encodes a constitutively active RalA unable to interact with the exocyst complex components Exoc2 and Exoc8, did not improve the process lengths of double-mutant SCs to control levels (Fig. 8 E). Similarly, we did not observe a rescue when expressing RalA72L A48W, a constitutively active RalA uncoupled from Exoc8 alone (Fig. 8 F). These results suggest that active RalA promotes process extension in SCs through interaction with the exocyst complex.

## Discussion

Radial sorting is a critical step in SC development that is necessary for these cells to enter into a one-to-one relationship with axons as a prerequisite for subsequent myelination. Our results establish that the small Ras-like GTPases RalA and RalB are required for the correct course of this process. By ablating expression of RalA in SCs and of RalB constitutively in all cell types, we demonstrate that loss of both Ral GTPases leads to a severe delay and partial block in radial axonal sorting. We further show that this phenotype is due to the combined loss of RalA and RalB functions specifically in SCs. Since Ral GTPases are involved in the regulation of proliferation (Kashatus et al., 2011), we analyzed SC proliferation and apoptosis. Instead of observing decreased proliferation, which might have explained the radial sorting defect present in RalA/B double-mutant nerves, we found an increased percentage of proliferating SCs

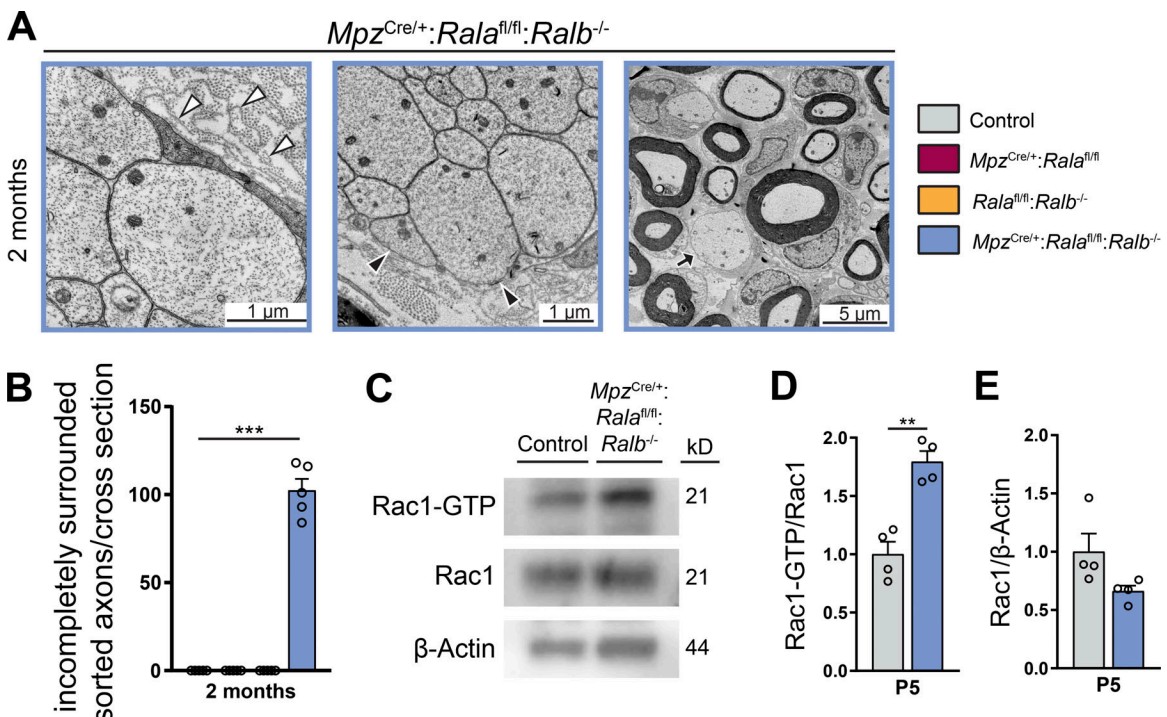

Figure 6. **Ral-deficient SCs show defective processes and redundant basal laminae in sciatic nerves. (A)** Exemplary high-magnification electron micrographs of sciatic nerves of 2-mo-old $Mpz^{Cre/+}$:$Rala^{fl/fl}$:$Ralb^{-/-}$ mice. In the left image, white arrowheads point to redundant loops of basal lamina associated with collagen. In the middle image, black arrowheads mark an area of a bundle of unsorted axons that is not covered by a SC process. In the right image, the arrow indicates an exemplary sorted but not-myelinated axon that is not fully surrounded by a SC. **(B)** Quantification of incompletely surrounded, not-myelinated sorted axons with a diameter >1 µm per sciatic nerve cross section in 2-mo-old mice. $n$ = 5 mice per genotype, with one complete cross section per animal analyzed. One-sample $t$ test. **(C–E)** Pull-down assay for active Rac1 from lysates of P5 sciatic nerves. An exemplary Western blot is shown in C. The levels of active Rac1 in D and total Rac1 in E were normalized to controls. $n$ = 4 samples per genotype, each containing sciatic nerves of three animals. Unpaired two-tailed $t$ test. All data are shown as mean ± SEM. **, $P < 0.01$; ***, $P < 0.001$.

without detectable changes in SC apoptosis. Closer examination of the SC morphology in double-mutant nerves pointed toward potential defects in the formation and maintenance of cellular processes, a feature that was confirmed by examining SCs ex vivo in culture. Through expression of previously described mutant RalA variants in cultured SCs, we were then able to determine that the interaction of RalA with the exocyst complex is necessary to promote cellular process extensions.

RalA and RalB are highly similar proteins, sharing 85% of their amino acid sequence and identical effector-binding regions (Chardin and Tavitian, 1989). They are thought to be regulated by the same set of GEFs and GAPs and use the same downstream effectors. Reports on functional redundancy and compensation between the two GTPases vary, however, depending on the experimental setting used. Constitutive RalB null mice are viable with no overt phenotype, while constitutive RalA deficiency leads to exencephaly and embryonic lethality (Peschard et al., 2012). In a mouse model of Kras-driven non-small cell lung carcinoma, expression of either RalA or RalB is sufficient to drive tumor growth (Peschard et al., 2012). Meanwhile, multiple studies reported that Ras-driven transformation depends on RalA and not RalB in various human cancer cell lines (Chien and White, 2003; Lim et al., 2005, 2006; Sablina et al., 2007). We found that expression of one Ral GTPase alone in SCs was sufficient to achieve normal radial sorting in peripheral nervous system development, indicating that Ral proteins are either functionally redundant here or can compensate for the loss of each other directly or by indirect effects. In this context, we detected a mild up-regulation of RalB mRNA upon loss of RalA, with no significant change at the protein levels. We have not followed up on this issue further. However, a possible compensation may also involve other mechanisms. There is evidence suggesting that divergent functions of the two Ral GTPases are mediated by distinct intracellular localization (Shipitsin and Feig, 2004; Falsetti et al., 2007). Thus, shifts in the localization of the remaining Ral GTPase could potentially be sufficient to compensate for loss of the other. In addition, activity levels of Ral proteins can be flexibly regulated by RalGEFs and RalGAPs, providing yet another mechanism by which SCs may be able to compensate for the loss of one Ral GTPase (Peschard et al., 2012).

Since we had identified axonal radial sorting as the main developmental process dependent on Ral proteins in SCs, we searched for the cellular mechanisms underlying these findings. Ultrastructural analysis of RalA/B double-mutant nerves revealed detached basal lamina and loops of redundant basal lamina as prominent features, together with sorted axons and bundles of unsorted axons that appeared to be covered only partially, or not at all, by SC processes. These observations are consistent with defects in the extension and stability of SC processes. In support of this interpretation, when RalA/B

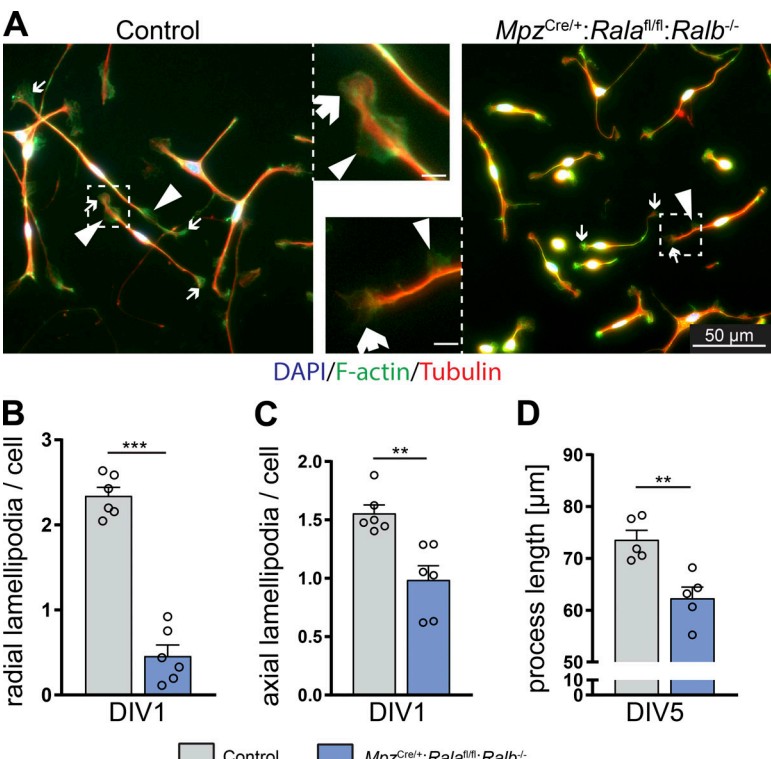

**A** Control *Mpz*^Cre/+^:*Rala*^fl/fl^:*Ralb*^-/-^

DAPI/F-actin/Tubulin

**B**
radial lamellipodia / cell
DIV1

**C**
axial lamellipodia / cell
DIV1

**D**
process length [μm]
DIV5

Control *Mpz*^Cre/+^:*Rala*^fl/fl^:*Ralb*^-/-^

Figure 7. **Cultured Ral-deficient SCs show process extension defects. (A)** SCs isolated from P5 sciatic nerves of control and *Mpz*^Cre/+^:*Rala*^fl/fl^:*Ralb*^−/−^ mice, grown in culture for 24 h on laminin 111. Processes and lamellipodia were visualized by immunostaining of α-tubulin (red) and F-actin (green), and nuclei were labeled with DAPI. Arrows point to examples of axial lamellipodia, and arrowheads point to radial lamellipodia. Scale bars of insets: 5 μm. **(B and C)** Number of radial (B) and axial (C) lamellipodia per SC from P5 sciatic nerves at DIV1, plated on laminin 111. *n* = cells derived from six individual mice per genotype; ≥100 cells analyzed per animal. Unpaired two-tailed *t* test. **(D)** Average process length of SCs from P5 sciatic nerves at DIV5, plated on laminin 111. *n* = cells derived from five individual mice per genotype; ≥100 cells analyzed per animal. Unpaired two-tailed *t* test. All data are shown as mean ± SEM. **, P < 0.01; ***, P < 0.001.

double-mutant SCs were taken into cell culture, we found that they formed shorter cell processes and fewer lamellipodia than controls. Similar morphological features have been described in mice lacking SC-expressed integrin β1 (Feltri et al., 2002) or its downstream signaling protein, Rac1 (Benninger et al., 2007; Nodari et al., 2007), consistent with a potential functional connection. Indeed, evidence for a link between Ral GTPases and integrin signaling has been provided in other settings. First, RalA mediates membrane raft exocytosis in response to integrin signaling through interaction with the exocyst complex in mouse embryonic fibroblasts (Balasubramanian et al., 2010). Second, RalA activity is increased when cortical neurons are plated on laminin (Lalli, 2009). Third, cytoskeletal dynamics are a well-known functional target of the signaling network downstream of integrin β1 (Feltri et al., 2016), in line with our findings of aberrant cell protrusions in Ral-deficient SCs. Furthermore, several studies involving reduction of signaling components downstream of integrins describe various degrees of impairments in SC protrusions (Benninger et al., 2007; Nodari et al., 2007; Pereira et al., 2009; Montani et al., 2014). Along these lines, we considered that lack of Ral GTPases in SCs may also have resulted in altered Rac1 activity. Intriguingly, we found increased Rac1 activity in P5 RalA/B double-mutant nerves. We cannot currently provide a definitive explanation for this finding and its potential relation to the phenotype of RalA/B double-mutant nerves. Molecularly, one might speculate that in mutants, reduced activity of the Ral effector and Rac1 GAP RalBP1 is involved (Cantor et al., 1995; Jullien-Flores et al., 1995; Park and Weinberg, 1995; Matsubara et al., 1997). However, whether the observed increase in Rac1 activity is relevant to the mutant phenotype remains to be determined. With regard to potential

functional contributions of RalBP1, our studies in cultured SCs did not support a major role of this protein in aiding Ral-mediated SC process extensions. Instead, we identified the exocyst complex as a mediator of Ral function. Although aberrant SC process extensions in cell culture are commonly correlated with axonal sorting and myelination defects in vivo (as in our study), we recognize that cell culture experiments have limitations with regard to comparisons to the in vivo setting. Thus, determination of the precise stages in SC myelination in which (and how) the Ral-exocyst connection is involved requires further investigations. Similarly, contributions of Ral effectors for which we did not find support for a regulatory role in SC process extension assays may still be involved in some aspects of SC biology.

Even though the currently available evidence appears as too fragmentary to provide a definitive direct link between the Ral-exocyst connection, integrin signaling, Rho GTPases, and SC development, it is noteworthy that data obtained from various angles suggest a conceivable convergence of the Rho GTPase signaling pathway with the exocyst complex and potentially Ral GTPases. The Ral effector Exoc2 can directly bind the RhoA-GEF GEF-H1 (Biondini et al., 2015). Moreover, Exoc4 and Exoc8 can interact with the Rac-GAP SH3BP1, and this interaction is important for the stability of the leading edge of cell processes (Parrini et al., 2011). Also, the Rac1-effector WAVE regulatory complex (WRC) can interact with the exocyst complex (Biondini et al., 2016). SH3BP1, WRC, and the exocyst complex can all be found at the leading edge of migratory cells (Parrini et al., 2011; Biondini et al., 2015), in line with a proposed model suggesting that the exocyst complex serves as a "molecular taxi" that transports signaling molecules to the sites of active process

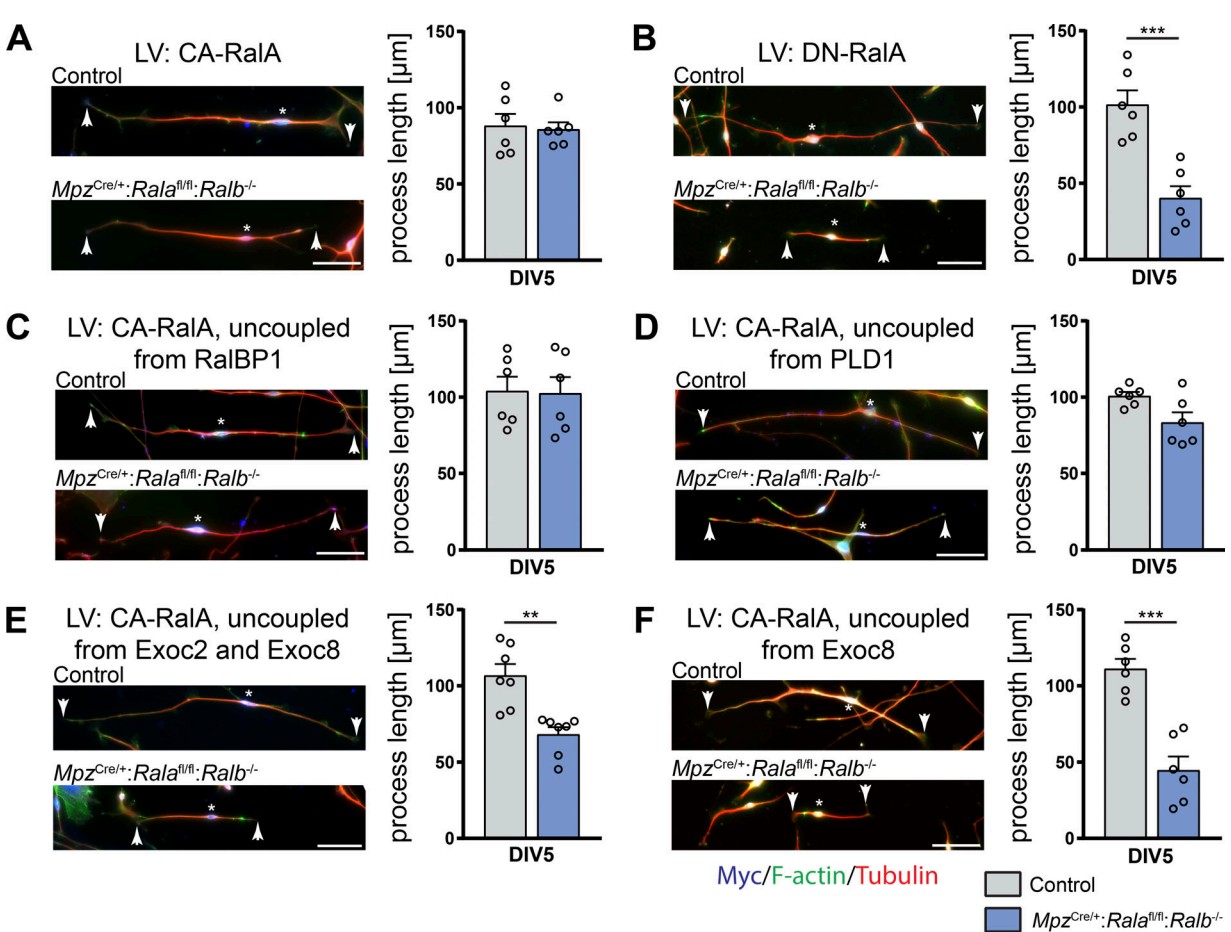

Figure 8. **Interaction of RalA with the exocyst complex is crucial for SC process extension.** Exemplary images of a control and a $Mpz^{Cre/+}:Rala^{fl/fl}:Ralb^{-/-}$ SC derived from P5 sciatic nerves infected with lentivirus (LV) expressing (A) myc-tagged RalA72L (constitutively active RalA [CA-RalA]), (B) myc-tagged RalA28N (dominant-negative RalA [DN-RalA]), (C) myc-tagged RalA72L D49N (CA-RalA uncoupled from the interaction with RalBP1), (D) myc-tagged RalA72L ΔN11 (CA-RalA uncoupled from the interaction with PLD1), (E) myc-tagged RalA72L D49E (CA-RalA uncoupled from the interaction with Exoc2 and Exoc8), and (F) myc-tagged RalA72L A48W (CA-RalA uncoupled from the interaction with Exoc8). SC processes were visualized by immunostaining of α-tubulin (red) and F-actin (green), and infection was controlled by immunostaining for myc (blue). Arrows indicate the far ends of SC processes, and asterisks mark the location of SC nuclei along the length of the cell. Quantifications of average process lengths are depicted on the right. $n$ = cells derived from seven (only in E) or six individual animals; ≥100 cells analyzed per animal. Multiple unpaired two-tailed $t$ tests with Holm–Sidak method for multiple comparisons correction. Scale bar for all images: 50 µm. All data are shown as mean ± SEM. **, $P < 0.01$; ***, $P < 0.001$.

extension (Zago et al., 2017, 2018). Furthermore, the actin nucleator complex Arp2/3 interacts with the WRC to regulate actin dynamics (Molinie and Gautreau, 2018), consistent with a critical role of Arp2/3 in oligodendrocyte process formation (Zuchero et al., 2015). Arp2/3 activity is also regulated by N-WASP (Molinie and Gautreau, 2018). Intriguingly, ablation of N-WASP expression in SCs caused defects in myelination and process extension but did not majorly impair radial sorting (Jin et al., 2011; Novak et al., 2011). These findings are broadly in agreement with the concept that Cdc42 and N-WASP are signaling on the Arp2/3 complex to regulate mainly filopodia formation, while Rac1 and the WRC activate Arp2/3 to promote the formation of lamellipodia (Molinie and Gautreau, 2018). This principle cross-talk warrants further investigations.

In adult RalA/B double-mutant mice, abnormal myelin profiles (including infoldings, outfoldings, and tomacula) were present in sciatic nerves at increased but rather low frequency. Such aberrations are reminiscent of the dysregulation of myelin

production that is characteristic of Mtmr2-deficient mice (Bolino et al., 2004; Bonneick et al., 2005; Cotter et al., 2010). Since interplay among Mtmr2, Discs large homolog 1, and exocyst complex member Exoc4 has been described to regulate membrane homeostasis in SC myelination (Bolis et al., 2009), these morphological observations are in line with a potential functional link between Ral proteins and other exocyst-related control elements of myelination. This hypothesis needs to be followed up in suitable experimental settings.

## Materials and methods
### Mice
Mice with floxed alleles for *Rala* ($Rala^{fl}$; RRID:IMSR_EM:10002) and null alleles for *Ralb* ($Ralb^-$; RRID:MGI:5505292) were generated previously (Peschard et al., 2012). These mice were on an FVBN background and subsequently crossed with *Mpz*-Cre mice on a C57B6 background (RRID:IMSR_JAX:017927; Feltri et al.,

1999). Experimental mice were of mixed background between the third and sixth generation of backcrosses into C57B6. For all experiments involving *Ralb⁻* alleles, age-matched mice from parallel breedings were used. Experimental genotypes are referred to as follows: $Mpz^{Cre/+}:Rala^{fl/fl}$, SC-specific deletion of RalA; $Rala^{fl/fl}:Ralb^{-/-}$, constitutive deletion of RalB; and $Mpz^{Cre/+}:Rala^{fl/fl}:Ralb^{-/-}$, SC-specific deletion of RalA on the background of constitutive RalB deletion. $Rala^{fl/fl}$ mice served as controls.

To generate SC-specific RalA/B double mutants, mice with floxed alleles for both *Rala* ($Rala^{fl}$) and *Ralb* ($Ralb^{fl}$; RRID:IMSR_EM:10003) were crossed with the *Mpz*-Cre driver line (Feltri et al., 1999). Experimental mice were on C57B6 background (>10 generations of backcrosses). For all experiments involving $Ralb^{fl}$ alleles, age-matched mice from parallel breedings were used. Experimental genotypes are referred to as follows: $Mpz^{Cre/+}:Rala^{fl/fl}$, SC-specific deletion of RalA; $Mpz^{Cre/+}:Ralb^{fl/fl}$, SC-specific deletion of RalB; and $Mpz^{Cre/+}:Rala^{fl/fl}:Ralb^{fl/fl}$, SC-specific deletion of RalA and RalB. Mice carrying floxed alleles for *Rala* ($Rala^{fl/fl}$), *Ralb* ($Ralb^{fl/fl}$), or both ($Rala^{fl/fl}:Ralb^{fl/fl}$) served as controls.

Genotypes were determined by genomic PCR with the following primers: Cre forward, 5′-ACCAGGTTCGTTCACTCATGG-3′; Cre reverse, 5′-AGGCTAAGTGCCTTCTCTACA-3′; RalA forward, 5′-GATGCCCTTAATGCAAATGACC-3′; RalA reverse, 5′-GCCATAGCAACGAGACAAGCC-3′; RalB forward, 5′-GGAGGCATGGGAAGATTAGAAG-3′; RalB null, 5′-GTCTGCTTACACACCTGTGTAC-3′; RalB reverse, 5′-CCCAAGCCAGAGATGCCTCAC-3′. All mice were cohoused in cages with a maximum of five mice, kept in a 12-h light and dark cycle, and fed standard chow ad libitum. Animals of either gender were used for the experiments. All animal experiments were approved by the Zurich Cantonal Veterinary Office and conducted in accordance with their guidelines.

## Motor behavior analysis (CatWalk)
For analysis of motor behavior of adult mice the CatWalk XT system (Noldus) was used. Mice were placed on the running field and left to traverse the field of their own accord. Per mouse, three compliant runs (run duration between 0.5 and 5 s, maximum allowed speed variation of 60%) were considered. Analysis was performed with CatWalk XT 10.6 software (Noldus). Stride length was measured on the left hindlimb, and base of support was measured on the hindpaws.

## Morphological analysis
To prepare for EM sectioning, sciatic nerves were fixed immediately after dissection with 3% glutaraldehyde and 4% paraformaldehyde in 0.1 M phosphate buffer. Nerves were incubated in 1% osmium tetroxide (EMS), dehydrated by serial incubations with increasing amounts of acetone, and embedded in Spurr resin (EMS). To obtain representative high-resolution micrographs of basal lamina and SC processes, ultrathin sections (65 nm) were imaged with a Morgagni 268 transmission electron microscope (Field Electron and Ion Company). To obtain reconstructions of the entire sciatic nerve, 99-nm-thick sections were collected on ITO coverslips (Optics Balzers) and imaged with either a Zeiss Gemini Leo 1530 FEG or a Zeiss Merlin FEG

scanning electron microscope attached to ATLAS modules. Image alignment and processing was performed with Photoshop CS6 or CC (RRID:SCR_014199; Adobe). To determine the numbers of myelinated, not-myelinated, and sorted fibers as well as the area of sciatic nerve occupied by bundles, the whole endoneurial area was analyzed. For g-ratio calculations, the axon diameter was derived from the axon area and the fiber diameter was obtained by adding twice the average myelin thickness measured at different locations. Per animal, ≥100 fibers derived from four randomly chosen different regions of the sciatic nerve were measured.

## Preparation and culture of primary mouse SCs
Sciatic nerves were isolated from P5 mice, the perineurium was removed, and nerves were digested with 1.25 mg/ml Trypsin (T9201; Sigma-Aldrich) and 2 mg/ml Collagenase (C0130; Sigma-Aldrich) in HBSS (Life Technologies) for 1 h. Cells were pelleted by centrifugation, resuspended in DMEM GlutaMAX +10% FCS (Life Technologies), and seeded in 24-well plates on coverslips coated with 20 µg/ml laminin 111 (L2020; Sigma-Aldrich). 16 h after seeding, the medium was changed to N2 SC medium (N2 supplement [Life Technologies], 10 ng/ml recombinant human EGF domain of neuregulin-1 β1 [R&D Systems], and 2.5 µM Forskolin [Sigma-Aldrich] in Advanced DMEM/F-12 [Life Technologies]). For analysis of lamellipodia, cells were fixed 24 h after plating. For virus infection experiments, 10 µl of concentrated virus was added per well to the N2 SC medium 16 h after plating and cells were fixed on DIV5. At the latter time point, a reliable quantification of lamellipodia was not possible due to the high density of the cultured cells that was required for survival. Labeling of infected cells by immunostaining for the myc tag allowed quantification of their process lengths. We did not observe adverse effects of viral infections on the SCs.

For all experiments, cells from individual animals were kept separate to provide biological replicates. To account for technical reproducibility, each experiment was repeated twice with cells from two to four animals at a time.

## Lentiviral vectors and virus production
The following myc-tagged RalA mutants were used (Lalli and Hall, 2005): RalA72L (constitutively active; Emkey et al., 1991), RalA28N (dominant negative; Jiang et al., 1995), RalA72L D49N (constitutively active, uncoupled from RalBP1; Cantor et al., 1995), RalA72L ΔN11 (constitutively active, uncoupled from PLD1; Jiang et al., 1995), RalA72L D49E (constitutively active, uncoupled from Exoc2 and Exoc8; Moskalenko et al., 2002, 2003), and RalA72L A48W (constitutively active, uncoupled from Exoc8; Cascone et al., 2008). All constructs were amplified by PCR, verified by sequencing, and inserted into pSicoR-Δ3′-loxP (modified version of pSicoR [Ventura et al., 2004] with a deleted 3′-loxP site) between the NheI and EcoRI restriction sites under control of the cytomegalovirus promotor.

For production of concentrated viruses, two 15-cm dishes of mycoplasma-free HEK293T cells (RRID:CVCL_0063) were transfected per construct with the lentiviral vector and the packaging plasmids psPAX2 and pCMV-VSV-G using Lipofectamine 2000 (Life Technologies) according to the manufacturer's

instructions. Supernatants were collected 72 h after transfection and filtered through a 45-μm sterile filter to remove cellular debris, and viruses were concentrated by ultracentrifugation at 21,000 rpm and 11°C for 2 h using a Sorvall WX 80+ ultracentrifuge and a Sorvall SureSpin 630 Swinging Bucket Rotor (Thermo Fisher Scientific). The pelleted viruses were resuspended in 500 μl PBS and used as indicated above. This resulted in the MOIs, as determined by QuickTiter Lentivirus Titer Kit (VPK-107; Cell Biolabs) according to the manufacturer's instructions: RalA72L = 1,541, RalA28N = 1,436, RalA72L D49N = 1,468, RalA72L ΔN11 = 1,631, RalA72L D49E = 1,496, and RalA72L A48W = 1,534.

### Antibodies
The following primary antibodies were used: RalA (610221, RRID:AB_397618, 1:1,000; BD Biosciences), RalB (MAB3920, RRID:AB_2176037, 1:1,000; R&D Systems), α-Tubulin (for immunoblot: T5168, RRID:AB_477579, 1:1,000; Sigma-Aldrich; for immunostaining: ab18251, RRID:AB_2210057, 1:500; Abcam), Sox10 (AF2864, RRID:AB_442208, 1:200; R&D Systems), CC3 (9664, RRID:AB_2070042, 1:500; Cell Signaling Technology), myc tag (ab32, RRID:AB_303599, 1:500; Abcam), and Rac1 (05–389, RRID:AB_309712; Millipore). Alexa Fluor 488–coupled Phalloidin (used 1:40) was purchased from Life Technologies. HRP- and fluorophore-coupled secondary antibodies (used 1:200 for immunostainings and 1:10,000 for immunoblots) were obtained from Life Technologies (Carlsbad) or Jackson ImmunoResearch.

### Immunostaining
Sciatic nerves were fixed with 4% paraformaldehyde in PBS for 1 h at 4°C, incubated for 1 h in 10% sucrose followed by overnight incubation in 20% sucrose at 4°C. Nerves were embedded in OCT (Tissue Tek), and 8-μm-thick sections were cut and stored at –80°C until further processing. Frozen slides were fixed for 10 min in 4% paraformaldehyde and permeabilized for 20 min in 0.25% Triton X-100 in PBS. Slides were blocked for 30 min in blocking buffer (1% BSA, 10% donkey serum, and 0.1% Triton X-100 in PBS), incubated overnight with primary antibodies, washed three times with PBS, incubated for 1 h with secondary antibodies, and counterstained with DAPI (Life Technologies). Finally, slides were mounted with Vectashield (Vector Laboratories). To analyze proliferation, P5 pups were injected with 50 μg per gram of body weight EdU (Life Technologies) 1 h before sacrificing. For EdU detection, the Click-iT EdU Alexa Fluor 647 kit (Life Technologies) was used according to the manufacturer's instructions.

To visualize the cytoskeleton of primary mouse SCs cultured on coverslips, cells were fixed with 4% paraformaldehyde in microtubule protection buffer (65 mM Pipes, 25 mM Hepes, 10 mM EGTA, and 3 mM MgCl$_2$, pH 6.9) for 10 min. Cells were permeabilized for 5 min with 0.1% Triton X-100 in PBS, incubated for 30 min with blocking buffer (10% goat serum and 1% BSA in PBS) and overnight with primary antibodies. Coverslips were incubated with secondary antibodies and Phalloidin–Alexa Fluor 488 (Life Technologies) for 1 h, counterstained with DAPI (Life Technologies), and mounted with ImmuMount (Thermo Fisher Scientific).

All Immunostainings were imaged with an Axio Imager.M2 (Zeiss) with a monochromatic charge-coupled device camera (sCMOS, pco.edge; PCO AG), a Plan Apochromat 10×/0.45 air objective (420640-9900; Zeiss), and Zen 2 software (Zeiss) at room temperature (20–24°C). An automatic stage was used to reconstruct full coverslips. For analysis of sciatic nerve sections, at least one representative section per animal was imaged and analyzed. To analyze lamellipodia of cultured mouse SCs, four to six representative fields per coverslip were imaged and lamellipodia were counted using Photoshop CC. We considered lamellipodia at the far ends of the main processes of a SC as axial and those along the length of the processes or SC cell body as radial. To determine process length of cultured mouse SCs, full coverslips were imaged and reconstructed. Process length was measured from the nucleus to the tip of the process using Fiji (version 2.0.0-rc-8/1.49c, RRID:SCR_002285; Schindelin et al., 2012; Schneider et al., 2012). Only cell processes originating directly from the cell body were considered as individual processes, and for each such process, the longest branch was measured. For each animal, two coverslips were imaged and analyzed. At least 100 cells per animal were considered. Primary mouse SCs had to be seeded at high density to ensure survival for 5 d in culture during viral infections. Therefore, for noninfected cells and lentiviral constructs with higher infection rates, two areas at the edge of each coverslip, where SCs are generally less dense, were selected, and ≥25 cells were measured per area. For lentiviral constructs with low infection rates, ≥50 infected cells per coverslip were randomly selected. For each animal, the lengths of all measured processes were averaged and depicted are the averages for each animal. To obtain representative images for all stainings the original images were false-colored, the individual channels were overlaid, and levels were adjusted using Photoshop CS6.

### RNA extraction
Sciatic nerves from P5 mice were extracted and placed in cold PBS. The perineurium was removed using two forceps, and then nerves were snap frozen in liquid nitrogen and stored at –80°C until further use. QIAzol (Qiagen) was used according to the manufacturer's instructions to isolate RNA for quantitative RT-PCR. RNA concentration was measured using a NanoDrop spectrophotometer (Thermo Fisher Scientific) and samples were stored at –80°C until further use.

### Reverse transcription and quantitative RT-PCR analysis
For reverse transcription, 190 ng total RNA was transcribed using the Maxima First Strand cDNA Synthesis Kit (K1641; Life Technologies) according to the manufacturer's instructions. Quantitative RT-PCR was performed using the FastStart Essential DNA Green Master (Roche) and Light Cycler 480 II (Roche). The following primers, all targeting mouse genes, were used: β-actin forward, 5′-GTCCACACCCGCCACC-3′; reverse, 5′-GGCCTCGTCACCCACATAG-3′; RalA forward, 5′-TTCCGAAGTGGGGAGGGATT-3′; reverse, 5′-TGCCTCTTCTACAGAAACCTGC-3′; RalB forward, 5′-GGTTGTGCGCATAGCCAGA-3′; reverse, 5′-GAAGCGTCAGGGCTGATTTG-3′. The results were quantified according to the 2$^{-\Delta\Delta Ct}$ method to obtain relative mRNA fold changes normalized to β-actin expression.

## Western blot

For protein analysis, sciatic nerves were extracted and immediately transferred to ice-cold PBS, and the perineurium was removed. Nerves were snap frozen in liquid nitrogen and stored at –80°C until further use. The frozen samples were mechanically disrupted using a small pestle and mixed with PN2 lysis buffer (25 mM Tris-HCl, pH 7.4, 95 mM NaCl, 10 mM EDTA, 2% SDS, and protease and phosphatase inhibitors; Roche). Samples were centrifuged for 15 min, and protein concentration was measured with a Micro BCA protein assay kit (Thermo Fisher Scientific) according to the manufacturer's instructions.

For SDS-PAGE, 10–15 µg protein was diluted with PN2 and 4× sample buffer (200 mM Tris-HCl, pH 6.8, 40% glycerol, 8% SDS, 20% β-mercaptoethanol, and 0.4% bromophenol blue). Samples were run on 4–15% polyacrylamide gradient gels (BioRad) and transferred onto polyvinylidene fluoride membranes (Millipore). Membranes were blocked for 1 h in 5% nonfat dry milk in TBS-T, incubated overnight at 4°C with primary antibodies diluted in 5% BSA in TBS-T, washed three times with TBS-T, and incubated for 1 h with HRP-conjugated secondary antibodies. Blots were incubated with ECL or ECL Prime (GE Healthcare) to produce chemiluminescent signals that were detected with Fusion FX7 (Vilber Lourmat). Densitometric quantification was performed with Fiji version 2.0.0-rc-8/1.49c (Schindelin et al., 2012; Schneider et al., 2012). For representative images, the levels were adjusted using Photoshop CS6. Apparent molecular weights were determined using Precision Plus Protein All Blue Standard (BioRad) or PageRuler Prestained Protein Ladder (Thermo Fisher Scientific).

## Rac1 activity assay

Rac1 activity assays were conducted as described previously (Sander et al., 1998; Benninger et al., 2007), using a GST-p21–activated kinase-crib domain construct provided by J. Collard (The Netherlands Cancer Institute, Amsterdam, Netherlands). Specifically, sciatic nerves of P5 control or mutant mice were extracted and immediately transferred to ice-cold PBS, and the perineurium was removed. For each sample, nerves from three mice per genotype were pooled, snap frozen in liquid nitrogen, and stored at –80°C until further use. The frozen samples were mechanically disrupted using a small pestle, homogenized in lysis buffer (10% glycerol, 50 mM Tris, pH 7.4, 100 mM NaCl, 1% NP-40, 2 mM MgCl$_2$, 0.005% Triton X-100, and protease and phosphatase inhibitors; Roche), and centrifuged for 5 min at 4°C. The total protein concentration of the supernatant was determined with a Micro BCA protein assay kit (Thermo Fisher Scientific) according to the manufacturer's instructions, and the protein concentrations of corresponding control and mutant samples were matched using lysis buffer. The obtained lysates were incubated with the bait protein bound to glutathione magnetic agarose beads (Thermo Fisher Scientific) for 45 min. Magnetic beads were washed three times with wash buffer (10% glycerol, 50 mM Tris, pH 7.4, 100 mM NaCl, 1% NP-40, 30 mM MgCl$_2$, and 0.005% Triton X-100), bound proteins were eluted using sample buffer (200 mM Tris-HCl, pH 6.8, 40% glycerol, 8% SDS, 20% β-mercaptoethanol, and 0.4% bromophenol blue), and samples were analyzed by Western blot as described above.

## Statistics

Statistical analyses were performed with GraphPad Prism (version 7.03, RRID:SCR_002798). Normal distribution and equal variances were assumed for all data but not formally tested due to the low number of replicates. Sample sizes were chosen in accordance to what is generally employed in the field. For all quantifications of microscopy images, the investigators were blinded to the genotype of the animals or cells. All data are shown as mean ± SEM. The number of biological replicates, the statistical test used for each figure, as well as mean, SEM, and exact P values are as follows.

*Figure 1*

**B.** $n$ = 4 (control), 5 (*Mpz*$^{Cre/+}$:*Rala*$^{fl/fl}$, *Mpz*$^{Cre/+}$:*Rala*$^{fl/fl}$:*Ralb*$^{-/-}$), or 6 (*Rala*$^{fl/fl}$:*Ralb*$^{-/-}$) mice per genotype; one-way ANOVA, Tukey's multiple comparisons test; mean ± SEM: control = 1.0 ± 0.045, *Mpz*$^{Cre/+}$:*Rala*$^{fl/fl}$ = 0.048 ± 0.007, *Rala*$^{fl/fl}$:*Ralb*$^{-/-}$ = 0.780 ± 0.106, *Mpz*$^{Cre/+}$:*Rala*$^{fl/fl}$:*Ralb*$^{-/-}$ = 0.089 ± 0.008; F$_{(3, 16)}$ = 48.72, P < 0.0001, P(control vs. *Mpz*$^{Cre/+}$:*Rala*$^{fl/fl}$) < 0.0001, P(control vs. *Rala*$^{fl/fl}$:*Ralb*$^{-/-}$) = 0.1494, P(control vs. *Mpz*$^{Cre/+}$:*Rala*$^{fl/fl}$:*Ralb*$^{-/-}$) < 0.0001, P(*Mpz*$^{Cre/+}$:*Rala*$^{fl/fl}$ vs. *Rala*$^{fl/fl}$:*Ralb*$^{-/-}$) < 0.0001, P(*Mpz*$^{Cre/+}$:*Rala*$^{fl/fl}$ vs. *Mpz*$^{Cre/+}$:*Rala*$^{fl/fl}$:*Ralb*$^{-/-}$) = 0.9737, P(*Rala*$^{fl/fl}$:*Ralb*$^{-/-}$ vs. *Mpz*$^{Cre/+}$:*Rala*$^{fl/fl}$:*Ralb*$^{-/-}$) < 0.0001.

**C.** $n$ = 5 animals per genotype; one-way ANOVA, Tukey's multiple comparisons test; mean ± SEM: control = 1.0 ± 0.121, *Mpz*$^{Cre/+}$:*Rala*$^{fl/fl}$ = 1.477 ± 0.163, *Rala*$^{fl/fl}$:*Ralb*$^{-/-}$ = 0.002 ± 0.001, *Mpz*$^{Cre/+}$:*Rala*$^{fl/fl}$:*Ralb*$^{-/-}$ = 0.005 ± 0.003; F$_{(3, 16)}$ = 52.88, P < 0.0001, P(control vs. *Mpz*$^{Cre/+}$:*Rala*$^{fl/fl}$) = 0.0204, P(control vs. *Rala*$^{fl/fl}$:*Ralb*$^{-/-}$) < 0.0001, P(control vs. *Mpz*$^{Cre/+}$:*Rala*$^{fl/fl}$:*Ralb*$^{-/-}$) < 0.0001, P(*Mpz*$^{Cre/+}$:*Rala*$^{fl/fl}$ vs. *Rala*$^{fl/fl}$:*Ralb*$^{-/-}$) < 0.0001, P(*Mpz*$^{Cre/+}$:*Rala*$^{fl/fl}$ vs. *Mpz*$^{Cre/+}$:*Rala*$^{fl/fl}$:*Ralb*$^{-/-}$) < 0.0001, P(*Rala*$^{fl/fl}$:*Ralb*$^{-/-}$ vs. *Mpz*$^{Cre/+}$:*Rala*$^{fl/fl}$:*Ralb*$^{-/-}$) > 0.9999.

**E.** $n$ = 6 animals per genotype; one-way ANOVA, Dunnett's multiple comparisons test; mean ± SEM: control = 1.0 ± 0.075, *Mpz*$^{Cre/+}$:*Rala*$^{fl/fl}$ = 0.501 ± 0.031, *Rala*$^{fl/fl}$:*Ralb*$^{-/-}$ = 1.14 ± 0.118, *Mpz*$^{Cre/+}$:*Rala*$^{fl/fl}$:*Ralb*$^{-/-}$ = 0.723 ± 0.048; F$_{(3, 20)}$ = 14.31, P < 0.0001, P(control vs. *Mpz*$^{Cre/+}$:*Rala*$^{fl/fl}$) = 0.0004, P(control vs. *Rala*$^{fl/fl}$:*Ralb*$^{-/-}$) = 0.4347, P(control vs. *Mpz*$^{Cre/+}$:*Rala*$^{fl/fl}$:*Ralb*$^{-/-}$) = 0.0446.

**F.** $n$ = 4 animals per genotype: one-way ANOVA, Dunnett's multiple comparisons test; mean ± SEM: control = 1.0 ± 0.279, *Mpz*$^{Cre/+}$:*Rala*$^{fl/fl}$ = 0.939 ± 0.222, *Rala*$^{fl/fl}$:*Ralb*$^{-/-}$ = 0.075 ± 0.013, *Mpz*$^{Cre/+}$:*Rala*$^{fl/fl}$:*Ralb*$^{-/-}$ = 0.051 ± 0.011; F$_{(3, 12)}$ = 8.648, P = 0.0025, P(control vs. *Mpz*$^{Cre/+}$:*Rala*$^{fl/fl}$) = 0.9898, P(control vs. *Rala*$^{fl/fl}$:*Ralb*$^{-/-}$) = 0.0085, P(control vs. *Mpz*$^{Cre/+}$:*Rala*$^{fl/fl}$:*Ralb*$^{-/-}$) = 0.0071.

**H.** $n$ = 9 animals (control) or 10 animals (*Mpz*$^{Cre/+}$:*Rala*$^{fl/fl}$, *Rala*$^{fl/fl}$:*Ralb*$^{-/-}$, *Mpz*$^{Cre/+}$:*Rala*$^{fl/fl}$:*Ralb*$^{-/-}$); one-way ANOVA, Tukey's multiple comparisons test; mean ± SEM: control = 2.866 ± 0.1, *Mpz*$^{Cre/+}$:*Rala*$^{fl/fl}$ = 2.982 ± 0.086, *Rala*$^{fl/fl}$:*Ralb*$^{-/-}$ = 2.652 ± 0.095, *Mpz*$^{Cre/+}$:*Rala*$^{fl/fl}$:*Ralb*$^{-/-}$ = 2.142 ± 0.153; F$_{(3, 35)}$ = 11.17, P < 0.0001, P(control vs. *Mpz*$^{Cre/+}$:*Rala*$^{fl/fl}$) = 0.8878, P(control vs. *Rala*$^{fl/fl}$:*Ralb*$^{-/-}$) = 0.5507, P(control vs. *Mpz*$^{Cre/+}$:*Rala*$^{fl/fl}$:*Ralb*$^{-/-}$) = 0.0004, P(*Mpz*$^{Cre/+}$:*Rala*$^{fl/fl}$ vs. *Rala*$^{fl/fl}$:*Ralb*$^{-/-}$) = 0.1706, P(*Mpz*$^{Cre/+}$:*Rala*$^{fl/fl}$ vs. *Mpz*$^{Cre/+}$:*Rala*$^{fl/fl}$:*Ralb*$^{-/-}$) < 0.0001, P(*Rala*$^{fl/fl}$:*Ralb*$^{-/-}$ vs. *Mpz*$^{Cre/+}$:*Rala*$^{fl/fl}$:*Ralb*$^{-/-}$) = 0.0126.

**I.** $n$ = 9 animals (control) or 10 animals ($Mpz^{Cre/+}$:$Rala^{fl/fl}$, $Rala^{fl/fl}$:$Ralb^{-/-}$, $Mpz^{Cre/+}$:$Rala^{fl/fl}$:$Ralb^{-/-}$); one-way ANOVA, Tukey's multiple comparisons test; mean ± SEM: control = 15.49 ± 2.457, $Mpz^{Cre/+}$:$Rala^{fl/fl}$ = 14.19 ± 3.338, $Rala^{fl/fl}$:$Ralb^{-/-}$ = 23.67 ± 3.777, $Mpz^{Cre/+}$:$Rala^{fl/fl}$:$Ralb^{-/-}$ = 35.04 ± 3.762; $F_{(3, 35)}$ = 7.931, P = 0.0004, P(control vs. $Mpz^{Cre/+}$:$Rala^{fl/fl}$) = 0.9934, P(control vs. $Rala^{fl/fl}$:$Ralb^{-/-}$) = 0.3562, P(control vs. $Mpz^{Cre/+}$:$Rala^{fl/fl}$:$Ralb^{-/-}$) = 0.0018, P($Mpz^{Cre/+}$:$Rala^{fl/fl}$ vs. $Rala^{fl/fl}$:$Ralb^{-/-}$) = 0.2134, P($Mpz^{Cre/+}$:$Rala^{fl/fl}$ vs. $Mpz^{Cre/+}$:$Rala^{fl/fl}$:$Ralb^{-/-}$) = 0.0006, P($Rala^{fl/fl}$:$Ralb^{-/-}$ vs. $Mpz^{Cre/+}$:$Rala^{fl/fl}$:$Ralb^{-/-}$) = 0.1.

**J.** $n$ = 9 animals (control) or 10 animals ($Mpz^{Cre/+}$:$Rala^{fl/fl}$, $Rala^{fl/fl}$:$Ralb^{-/-}$, $Mpz^{Cre/+}$:$Rala^{fl/fl}$:$Ralb^{-/-}$); one-way ANOVA, Tukey's multiple comparisons test; mean ± SEM: control = 9.043 ± 0.258, $Mpz^{Cre/+}$:$Rala^{fl/fl}$ = 9.483 ± 0.249, $Rala^{fl/fl}$:$Ralb^{-/-}$ = 8.792 ± 0.211, $Mpz^{Cre/+}$:$Rala^{fl/fl}$:$Ralb^{-/-}$ = 7.904 ± 0.209; $F_{(3, 35)}$ = 8.432, P = 0.0002, P(control vs. $Mpz^{Cre/+}$:$Rala^{fl/fl}$) = 0.5527, P(control vs. $Rala^{fl/fl}$:$Ralb^{-/-}$) = 0.8747, P(control vs. $Mpz^{Cre/+}$:$Rala^{fl/fl}$:$Ralb^{-/-}$) = 0.0082, P($Mpz^{Cre/+}$:$Rala^{fl/fl}$ vs. $Rala^{fl/fl}$:$Ralb^{-/-}$) = 0.1615, P($Mpz^{Cre/+}$:$Rala^{fl/fl}$ vs. $Mpz^{Cre/+}$:$Rala^{fl/fl}$:$Ralb^{-/-}$) = 0.0001, P($Rala^{fl/fl}$:$Ralb^{-/-}$ vs. $Mpz^{Cre/+}$:$Rala^{fl/fl}$:$Ralb^{-/-}$) = 0.0446.

## Figure 2

**B.** $n$ = 4 animals per genotype; one-way ANOVA, Tukey's multiple comparisons test; mean ± SEM: control = 1,611 ± 210.6, $Mpz^{Cre/+}$:$Rala^{fl/fl}$ = 2,061 ± 140.1, $Rala^{fl/fl}$:$Ralb^{-/-}$ = 1,541 ± 113.7, $Mpz^{Cre/+}$:$Rala^{fl/fl}$:$Ralb^{-/-}$ = 196.8 ± 68.73; $F_{(3, 12)}$ = 31.68, P < 0.0001, P(control vs. $Mpz^{Cre/+}$:$Rala^{fl/fl}$) = 0.1713, P(control vs. $Rala^{fl/fl}$:$Ralb^{-/-}$) = 0.9849, P(control vs. $Mpz^{Cre/+}$:$Rala^{fl/fl}$:$Ralb^{-/-}$) < 0.0001, P($Mpz^{Cre/+}$:$Rala^{fl/fl}$ vs. $Rala^{fl/fl}$:$Ralb^{-/-}$) = 0.0978, P($Mpz^{Cre/+}$:$Rala^{fl/fl}$ vs. $Mpz^{Cre/+}$:$Rala^{fl/fl}$:$Ralb^{-/-}$) < 0.0001, P($Rala^{fl/fl}$:$Ralb^{-/-}$ vs. $Mpz^{Cre/+}$:$Rala^{fl/fl}$:$Ralb^{-/-}$) = 0.0001.

**C.** $n$ = 4 animals per genotype; one-way ANOVA, Tukey's multiple comparisons test; mean ± SEM: control = 2,612 ± 251, $Mpz^{Cre/+}$:$Rala^{fl/fl}$ = 3,159 ± 82.61, $Rala^{fl/fl}$:$Ralb^{-/-}$ = 2,863 ± 177.5, $Mpz^{Cre/+}$:$Rala^{fl/fl}$:$Ralb^{-/-}$ = 1,260 ± 279.2; $F_{(3, 12)}$ = 15.72, P = 0.0002, P(control vs. $Mpz^{Cre/+}$:$Rala^{fl/fl}$) = 0.3083, P(control vs. $Rala^{fl/fl}$:$Ralb^{-/-}$) = 0.8346, P(control vs. $Mpz^{Cre/+}$:$Rala^{fl/fl}$:$Ralb^{-/-}$) = 0.0034, P($Mpz^{Cre/+}$:$Rala^{fl/fl}$ vs. $Rala^{fl/fl}$:$Ralb^{-/-}$) = 0.7594, P($Mpz^{Cre/+}$:$Rala^{fl/fl}$ vs. $Mpz^{Cre/+}$:$Rala^{fl/fl}$:$Ralb^{-/-}$) = 0.0002, P($Rala^{fl/fl}$:$Ralb^{-/-}$ vs. $Mpz^{Cre/+}$:$Rala^{fl/fl}$:$Ralb^{-/-}$) = 0.0009.

**D.** $n$ = 4 animals per genotype; one-way ANOVA, Tukey's multiple comparisons test; mean ± SEM: control = 1,001 ± 98.53, $Mpz^{Cre/+}$:$Rala^{fl/fl}$ = 1,098 ± 146.2, $Rala^{fl/fl}$:$Ralb^{-/-}$ = 1,323 ± 163.9, $Mpz^{Cre/+}$:$Rala^{fl/fl}$:$Ralb^{-/-}$ = 1,063 ± 210.8; $F_{(3, 12)}$ = 0.7678, P = 0.5338, P(control vs. $Mpz^{Cre/+}$:$Rala^{fl/fl}$) = 0.9722, P(control vs. $Rala^{fl/fl}$:$Ralb^{-/-}$) = 0.5102, P(control vs. $Mpz^{Cre/+}$:$Rala^{fl/fl}$:$Ralb^{-/-}$) = 0.9923, P($Mpz^{Cre/+}$:$Rala^{fl/fl}$ vs. $Rala^{fl/fl}$:$Ralb^{-/-}$) = 0.7565, P($Mpz^{Cre/+}$:$Rala^{fl/fl}$ vs. $Mpz^{Cre/+}$:$Rala^{fl/fl}$:$Ralb^{-/-}$) = 0.9986, P($Rala^{fl/fl}$:$Ralb^{-/-}$ vs. $Mpz^{Cre/+}$:$Rala^{fl/fl}$:$Ralb^{-/-}$) = 0.6692.

**E.** $n$ = 4 animals per genotype; one-way ANOVA, Tukey's multiple comparisons test; mean ± SEM: control = 4.55 ± 0.46, $Mpz^{Cre/+}$:$Rala^{fl/fl}$ = 4.64 ± 0.42, $Rala^{fl/fl}$:$Ralb^{-/-}$ = 5.88 ± 0.66, $Mpz^{Cre/+}$:$Rala^{fl/fl}$:$Ralb^{-/-}$ = 18.21 ± 2.94; $F_{(3, 12)}$ = 18.53, P < 0.0001, P(control vs. $Mpz^{Cre/+}$:$Rala^{fl/fl}$) > 0.9999, P(control vs. $Rala^{fl/fl}$:$Ralb^{-/-}$) = 0.9258, P(control vs. $Mpz^{Cre/+}$:$Rala^{fl/fl}$:$Ralb^{-/-}$) = 0.0002, P($Mpz^{Cre/+}$:$Rala^{fl/fl}$ vs. $Rala^{fl/fl}$:$Ralb^{-/-}$) = 0.9385, P($Mpz^{Cre/+}$:$Rala^{fl/fl}$

vs. $Mpz^{Cre/+}$:$Rala^{fl/fl}$:$Ralb^{-/-}$) = 0.0002, P($Rala^{fl/fl}$:$Ralb^{-/-}$ vs. $Mpz^{Cre/+}$:$Rala^{fl/fl}$:$Ralb^{-/-}$) = 0.0005.

**F.** $n$ = 5 animals per genotype; one-way ANOVA, Tukey's multiple comparisons test; mean ± SEM: control = 3,767 ± 111.4, $Mpz^{Cre/+}$:$Rala^{fl/fl}$ = 3,628 ± 143.1, $Rala^{fl/fl}$:$Ralb^{-/-}$ = 3,590 ± 127.6, $Mpz^{Cre/+}$:$Rala^{fl/fl}$:$Ralb^{-/-}$ = 2,624 ± 270.1; $F_{(3, 16)}$ = 9.004, P = 0.001, P(control vs. $Mpz^{Cre/+}$:$Rala^{fl/fl}$) = 0.9417, P(control vs. $Rala^{fl/fl}$:$Ralb^{-/-}$) = 0.8905, P(control vs. $Mpz^{Cre/+}$:$Rala^{fl/fl}$:$Ralb^{-/-}$) = 0.0014, P($Mpz^{Cre/+}$:$Rala^{fl/fl}$ vs. $Rala^{fl/fl}$:$Ralb^{-/-}$) = 0.9987, P($Mpz^{Cre/+}$:$Rala^{fl/fl}$ vs. $Mpz^{Cre/+}$:$Rala^{fl/fl}$:$Ralb^{-/-}$) = 0.0045, P($Rala^{fl/fl}$:$Ralb^{-/-}$ vs. $Mpz^{Cre/+}$:$Rala^{fl/fl}$:$Ralb^{-/-}$) = 0.0061.

**G.** $n$ = 5 animals per genotype; one-way ANOVA, Tukey's multiple comparisons test; mean ± SEM: control = 3,769 ± 111.8, $Mpz^{Cre/+}$:$Rala^{fl/fl}$ = 3,632 ± 144.1, $Rala^{fl/fl}$:$Ralb^{-/-}$ = 3,601 ± 129, $Mpz^{Cre/+}$:$Rala^{fl/fl}$:$Ralb^{-/-}$ = 2,798 ± 271.7; $F_{(3, 16)}$ = 6.271, P = 0.0051, P(control vs. $Mpz^{Cre/+}$:$Rala^{fl/fl}$) = 0.9459, P(control vs. $Rala^{fl/fl}$:$Ralb^{-/-}$) = 0.9049, P(control vs. $Mpz^{Cre/+}$:$Rala^{fl/fl}$:$Ralb^{-/-}$) = 0.0063, P($Mpz^{Cre/+}$:$Rala^{fl/fl}$ vs. $Rala^{fl/fl}$:$Ralb^{-/-}$) = 0.9992, P($Mpz^{Cre/+}$:$Rala^{fl/fl}$ vs. $Mpz^{Cre/+}$:$Rala^{fl/fl}$:$Ralb^{-/-}$) = 0.0190, P($Rala^{fl/fl}$:$Ralb^{-/-}$ vs. $Mpz^{Cre/+}$:$Rala^{fl/fl}$:$Ralb^{-/-}$) = 0.0245.

**H.** $n$ = 5 animals per genotype; one-way ANOVA, Tukey's multiple comparisons test; mean ± SEM: control = 2 ± 1.304, $Mpz^{Cre/+}$:$Rala^{fl/fl}$ = 4.8 ± 2.035, $Rala^{fl/fl}$:$Ralb^{-/-}$ = 10.2 ± 3.089, $Mpz^{Cre/+}$:$Rala^{fl/fl}$:$Ralb^{-/-}$ = 174.6 ± 20.02; $F_{(3, 16)}$ = 68.66, P < 0.0001, P(control vs. $Mpz^{Cre/+}$:$Rala^{fl/fl}$) = 0.9973, P(control vs. $Rala^{fl/fl}$:$Ralb^{-/-}$) = 0.9401, P(control vs. $Mpz^{Cre/+}$:$Rala^{fl/fl}$:$Ralb^{-/-}$) < 0.0001, P($Mpz^{Cre/+}$:$Rala^{fl/fl}$ vs. $Rala^{fl/fl}$:$Ralb^{-/-}$) = 0.9815, P($Mpz^{Cre/+}$:$Rala^{fl/fl}$ vs. $Mpz^{Cre/+}$:$Rala^{fl/fl}$:$Ralb^{-/-}$) < 0.0001, P($Rala^{fl/fl}$:$Ralb^{-/-}$ vs. $Mpz^{Cre/+}$:$Rala^{fl/fl}$:$Ralb^{-/-}$) < 0.0001.

**I.** $n$ = 5 animals per genotype; one-way ANOVA, Tukey's multiple comparisons test; mean ± SEM: control = 0.0 ± 0.0, $Mpz^{Cre/+}$:$Rala^{fl/fl}$ = 0.0 ± 0.0, $Rala^{fl/fl}$:$Ralb^{-/-}$ = 0.0 ± 0.0, $Mpz^{Cre/+}$:$Rala^{fl/fl}$:$Ralb^{-/-}$ = 2.68 ± 1.116; $F_{(3, 16)}$ = 5.763, P = 0.0072, P(control vs. $Mpz^{Cre/+}$:$Rala^{fl/fl}$) > 0.9999, P(control vs. $Rala^{fl/fl}$:$Ralb^{-/-}$) > 0.9999, P(control vs. $Mpz^{Cre/+}$:$Rala^{fl/fl}$:$Ralb^{-/-}$) = 0.0175, P($Mpz^{Cre/+}$:$Rala^{fl/fl}$ vs. $Rala^{fl/fl}$:$Ralb^{-/-}$) > 0.9999, P($Mpz^{Cre/+}$:$Rala^{fl/fl}$ vs. $Mpz^{Cre/+}$:$Rala^{fl/fl}$:$Ralb^{-/-}$) = 0.0175, P($Rala^{fl/fl}$:$Ralb^{-/-}$ vs. $Mpz^{Cre/+}$:$Rala^{fl/fl}$:$Ralb^{-/-}$) = 0.0175.

## Figure 3

**B.** $n$ = 5 animals (control, $Rala^{fl/fl}$:$Ralb^{-/-}$, $Mpz^{Cre/+}$:$Rala^{fl/fl}$:$Ralb^{-/-}$ at 2 mo and 1 yr; and $Mpz^{Cre/+}$:$Rala^{fl/fl}$ at 2 mo) or 4 animals ($Mpz^{Cre/+}$:$Rala^{fl/fl}$ at 1 yr) per genotype, two-way ANOVA, Tukey's multiple comparisons test; mean ± SEM: control$_{2\,mo}$ = 3,767 ± 111.4, $Mpz^{Cre/+}$:$Rala^{fl/fl}$$_{2\,mo}$ = 3,628 ± 143.1, $Rala^{fl/fl}$:$Ralb^{-/-}$$_{2\,mo}$ = 3,590 ± 127.6, $Mpz^{Cre/+}$:$Rala^{fl/fl}$:$Ralb^{-/-}$$_{2\,mo}$ = 2,624 ± 270.1, control$_{1\,yr}$ = 3,762 ± 127.6, $Mpz^{Cre/+}$:$Rala^{fl/fl}$$_{1\,yr}$ = 3,449 ± 154.5, $Rala^{fl/fl}$:$Ralb^{-/-}$$_{1\,yr}$ = 3,398 ± 88.73, $Mpz^{Cre/+}$:$Rala^{fl/fl}$:$Ralb^{-/-}$$_{1\,yr}$ = 1,495 ± 105.5; $F_{(3, 31)genotype}$ = 54.36, P(genotype) < 0.0001, $F_{(1, 31)age}$ = 12.36, P(age) = 0.0014; P(control$_{2\,mo}$ vs. $Mpz^{Cre/+}$:$Rala^{fl/fl}$$_{2\,mo}$) = 0.9975, P(control$_{2\,mo}$ vs. $Rala^{fl/fl}$:$Ralb^{-/-}$$_{2\,mo}$) = 0.9894, P(control$_{2\,mo}$ vs. $Mpz^{Cre/+}$:$Rala^{fl/fl}$:$Ralb^{-/-}$$_{2\,mo}$) = 0.0002, P($Mpz^{Cre/+}$:$Rala^{fl/fl}$$_{2\,mo}$ vs. $Rala^{fl/fl}$:$Ralb^{-/-}$$_{2\,mo}$) > 0.9999, P($Mpz^{Cre/+}$:$Rala^{fl/fl}$$_{2\,mo}$ vs. $Mpz^{Cre/+}$:$Rala^{fl/fl}$:$Ralb^{-/-}$$_{2\,mo}$) = 0.001, P($Rala^{fl/fl}$:$Ralb^{-/-}$$_{2\,mo}$ vs. $Mpz^{Cre/+}$:$Rala^{fl/fl}$:$Ralb^{-/-}$$_{2\,mo}$) = 0.0016; P(control$_{1\,yr}$ vs. $Mpz^{Cre/+}$:$Rala^{fl/fl}$$_{1\,yr}$) = 0.8501, P(control$_{1\,yr}$ vs. $Rala^{fl/fl}$:$Ralb^{-/-}$$_{1\,yr}$) =

0.6699, P(control$_{1\ yr}$ vs. $Mpz^{Cre/+}$:$Rala^{fl/fl}$:$Ralb^{-/-}{}_{1\ yr}$) < 0.0001, P($Mpz^{Cre/+}$:$Rala^{fl/fl}{}_{1\ yr}$ vs. $Ralb^{fl/fl}$:$Ralb^{-/-}{}_{1\ yr}$) > 0.9999, P($Mpz^{Cre/+}$:$Rala^{fl/fl}{}_{1\ yr}$ vs. $Mpz^{Cre/+}$:$Rala^{fl/fl}$:$Ralb^{-/-}{}_{1\ yr}$) < 0.0001, P($Rala^{fl/fl}$:$Ralb^{-/-}{}_{1\ yr}$ vs. $Mpz^{Cre/+}$:$Rala^{fl/fl}$:$Ralb^{-/-}{}_{1\ yr}$) < 0.0001; P(control$_{2\ mo}$ vs. control$_{1\ yr}$) > 0.9999, P($Mpz^{Cre/+}$:$Rala^{fl/fl}{}_{2\ mo}$ vs. $Mpz^{Cre/+}$:$Rala^{fl/fl}{}_{1\ yr}$) = 0.992, P($Rala^{fl/fl}$:$Ralb^{-/-}{}_{2\ mo}$ vs. $Rala^{fl/fl}$:$Ralb^{-/-}{}_{1\ yr}$) = 0.9825, P($Mpz^{Cre/+}$:$Rala^{fl/fl}$:$Ralb^{-/-}{}_{2\ mo}$ vs. $Mpz^{Cre/+}$:$Rala^{fl/fl}$:$Ralb^{-/-}{}_{1\ yr}$) = 0.0002.

**C.** $n$ = 5 animals (control, $Rala^{fl/fl}$:$Ralb^{-/-}$, $Mpz^{Cre/+}$:$Rala^{fl/fl}$:$Ralb^{-/-}$ at 2 mo and 1 yr; and $Mpz^{Cre/+}$:$Rala^{fl/fl}$ at 2 mo) or 4 animals ($Mpz^{Cre/+}$:$Rala^{fl/fl}$ at 1 yr) per genotype, two-way ANOVA, Tukey's multiple comparisons test; mean ± SEM: control$_{2\ mo}$ = 3,769 ± 111.8, $Mpz^{Cre/+}$:$Rala^{fl/fl}{}_{2\ mo}$ = 3,632 ± 144.1, $Rala^{fl/fl}$:$Ralb^{-/-}{}_{2\ mo}$ = 3,601 ± 129, $Mpz^{Cre/+}$:$Rala^{fl/fl}$:$Ralb^{-/-}{}_{2\ mo}$ = 2,798 ± 271.7, control$_{1\ yr}$ = 3,785 ± 128.9, $Mpz^{Cre/+}$:$Rala^{fl/fl}{}_{1\ yr}$ = 3,478 ± 152.1, $Rala^{fl/fl}$:$Ralb^{-/-}{}_{1\ yr}$ = 3,422 ± 87.22, $Mpz^{Cre/+}$:$Rala^{fl/fl}$:$Ralb^{-/-}{}_{1\ yr}$ = 1,875 ± 140.8; $F_{(3,\ 31)genotype}$ = 35.75, P(genotype) < 0.0001, $F_{(1,\ 31)age}$ = 7.941, P(age) = 0.0083; P(control$_{2\ mo}$ vs. $Mpz^{Cre/+}$:$Rala^{fl/fl}{}_{2\ mo}$) = 0.9981, P(control$_{2\ mo}$ vs. $Rala^{fl/fl}$:$Ralb^{-/-}{}_{2\ mo}$) = 0.9932, P(control$_{2\ mo}$ vs. $Mpz^{Cre/+}$:$Rala^{fl/fl}$:$Ralb^{-/-}{}_{2\ mo}$) = 0.0021, P($Mpz^{Cre/+}$:$Rala^{fl/fl}{}_{2\ mo}$ vs. $Rala^{fl/fl}$:$Ralb^{-/-}{}_{2\ mo}$) > 0.9999, P($Mpz^{Cre/+}$:$Rala^{fl/fl}{}_{2\ mo}$ vs. $Mpz^{Cre/+}$:$Rala^{fl/fl}$:$Ralb^{-/-}{}_{2\ mo}$) = 0.0114, P($Rala^{fl/fl}$:$Ralb^{-/-}{}_{2\ mo}$ vs. $Mpz^{Cre/+}$:$Rala^{fl/fl}$:$Ralb^{-/-}{}_{2\ mo}$) = 0.0165; P(control$_{1\ yr}$ vs. $Mpz^{Cre/+}$:$Rala^{fl/fl}{}_{1\ yr}$) = 0.8764, P(control$_{1\ yr}$ vs. $Rala^{fl/fl}$:$Ralb^{-/-}{}_{1\ yr}$) = 0.7009, P(control$_{1\ yr}$ vs. $Mpz^{Cre/+}$:$Rala^{fl/fl}$:$Ralb^{-/-}{}_{1\ yr}$) < 0.0001, P($Mpz^{Cre/+}$:$Rala^{fl/fl}{}_{1\ yr}$ vs. $Rala^{fl/fl}$:$Ralb^{-/-}{}_{1\ yr}$) > 0.9999, P($Mpz^{Cre/+}$:$Rala^{fl/fl}{}_{1\ yr}$ vs. $Mpz^{Cre/+}$:$Rala^{fl/fl}$:$Ralb^{-/-}{}_{1\ yr}$) < 0.0001, P($Rala^{fl/fl}$:$Ralb^{-/-}{}_{1\ yr}$ vs. $Mpz^{Cre/+}$:$Rala^{fl/fl}$:$Ralb^{-/-}{}_{1\ yr}$) < 0.0001; P(control$_{2\ mo}$ vs. control$_{1\ yr}$) > 0.9999, P($Mpz^{Cre/+}$:$Rala^{fl/fl}{}_{2\ mo}$ vs. $Mpz^{Cre/+}$:$Rala^{fl/fl}{}_{1\ yr}$) = 0.9972, P($Rala^{fl/fl}$:$Ralb^{-/-}{}_{2\ mo}$ vs. $Rala^{fl/fl}$:$Ralb^{-/-}{}_{1\ yr}$) = 0.9903, P($Mpz^{Cre/+}$:$Rala^{fl/fl}$:$Ralb^{-/-}{}_{2\ mo}$ vs. $Mpz^{Cre/+}$:$Rala^{fl/fl}$:$Ralb^{-/-}{}_{1\ yr}$) = 0.0039.

*Figure 4*

**B.** $n$ = 5 animals per genotype; one-way ANOVA, Tukey's multiple comparisons test; mean ± SEM: control = 2,599 ± 67.55, $Mpz^{Cre/+}$:$Rala^{fl/fl}$ = 2,304 ± 186.9, $Mpz^{Cre/+}$:$Ralb^{fl/fl}$ = 2,493 ± 50.49, $Mpz^{Cre/+}$:$Rala^{fl/fl}$:$Ralb^{fl/fl}$ = 349.6 ± 54.96; $F_{(3,\ 16)}$ = 100.7, P < 0.0001, P(control vs. $Mpz^{Cre/+}$:$Rala^{fl/fl}$) = 0.2435, P(control vs. $Mpz^{Cre/+}$:$Ralb^{fl/fl}$) = 0.8952, P(control vs. $Mpz^{Cre/+}$:$Rala^{fl/fl}$:$Ralb^{fl/fl}$) < 0.0001, P($Mpz^{Cre/+}$:$Rala^{fl/fl}$ vs. $Mpz^{Cre/+}$:$Ralb^{fl/fl}$) = 0.6003, P($Mpz^{Cre/+}$:$Rala^{fl/fl}$ vs. $Mpz^{Cre/+}$:$Rala^{fl/fl}$:$Ralb^{fl/fl}$) < 0.0001, P($Mpz^{Cre/+}$:$Ralb^{fl/fl}$ vs. $Mpz^{Cre/+}$:$Rala^{fl/fl}$:$Ralb^{fl/fl}$) < 0.0001.

**C.** $n$ = 5 animals per genotype; one-way ANOVA, Tukey's multiple comparisons test; mean ± SEM: control = 3,293 ± 78.16, $Mpz^{Cre/+}$:$Rala^{fl/fl}$ = 3,255 ± 93.5, $Mpz^{Cre/+}$:$Ralb^{fl/fl}$ = 3,206 ± 54.21, $Mpz^{Cre/+}$:$Rala^{fl/fl}$:$Ralb^{fl/fl}$ = 1,090 ± 177.7; $F_{(3,\ 16)}$ = 94.65, P < 0.0001, P(control vs. $Mpz^{Cre/+}$:$Rala^{fl/fl}$) = 0.9948, P(control vs. $Mpz^{Cre/+}$:$Ralb^{fl/fl}$) = 0.9445, P(control vs. $Mpz^{Cre/+}$:$Rala^{fl/fl}$:$Ralb^{fl/fl}$) < 0.0001, P($Mpz^{Cre/+}$:$Rala^{fl/fl}$ vs. $Mpz^{Cre/+}$:$Ralb^{fl/fl}$) = 0.9892, P($Mpz^{Cre/+}$:$Rala^{fl/fl}$ vs. $Mpz^{Cre/+}$:$Rala^{fl/fl}$:$Ralb^{fl/fl}$) < 0.0001, P($Mpz^{Cre/+}$:$Ralb^{fl/fl}$ vs. $Mpz^{Cre/+}$:$Rala^{fl/fl}$:$Ralb^{fl/fl}$) < 0.0001.

**D.** $n$ = 5 animals per genotype; one-way ANOVA, Tukey's multiple comparisons test; mean ± SEM: control = 694 ± 60.83, $Mpz^{Cre/+}$:$Rala^{fl/fl}$ = 905.6 ± 88.37, $Mpz^{Cre/+}$:$Ralb^{fl/fl}$ = 677.8 ± 45.93, $Mpz^{Cre/+}$:$Rala^{fl/fl}$:$Ralb^{fl/fl}$ = 740.6 ± 125; $F_{(3,\ 16)}$ = 1.484, P = 0.2565, P(control vs. $Mpz^{Cre/+}$:$Rala^{fl/fl}$) = 0.332, P(control vs. $Mpz^{Cre/+}$:$Ralb^{fl/fl}$) = 0.9991, P(control vs. $Mpz^{Cre/+}$:$Rala^{fl/fl}$:$Ralb^{fl/fl}$) = 0.9799, P($Mpz^{Cre/+}$:$Rala^{fl/fl}$ vs. $Mpz^{Cre/+}$:$Ralb^{fl/fl}$) = 0.2735, P($Mpz^{Cre/+}$:$Rala^{fl/fl}$ vs. $Mpz^{Cre/+}$:$Rala^{fl/fl}$:$Ralb^{fl/fl}$) = 0.5380, P($Mpz^{Cre/+}$:$Ralb^{fl/fl}$ vs. $Mpz^{Cre/+}$:$Rala^{fl/fl}$:$Ralb^{fl/fl}$) = 0.9532.

**E.** $n$ = 5 animals per genotype; one-way ANOVA, Tukey's multiple comparisons test; mean ± SEM: control = 7.61 ± 0.297, $Mpz^{Cre/+}$:$Rala^{fl/fl}$ = 6.021 ± 0.603, $Mpz^{Cre/+}$:$Ralb^{fl/fl}$ = 6.377 ± 0.449, $Mpz^{Cre/+}$:$Rala^{fl/fl}$:$Ralb^{fl/fl}$ = 26.37 ± 1.992; $F_{(3,\ 16)}$ = 84.36, P < 0.0001, P(control vs. $Mpz^{Cre/+}$:$Rala^{fl/fl}$) = 0.7263, P(control vs. $Mpz^{Cre/+}$:$Ralb^{fl/fl}$) = 0.8484, P(control vs. $Mpz^{Cre/+}$:$Rala^{fl/fl}$:$Ralb^{fl/fl}$) < 0.0001, P($Mpz^{Cre/+}$:$Rala^{fl/fl}$ vs. $Mpz^{Cre/+}$:$Ralb^{fl/fl}$) = 0.9953, P($Mpz^{Cre/+}$:$Rala^{fl/fl}$ vs. $Mpz^{Cre/+}$:$Rala^{fl/fl}$:$Ralb^{fl/fl}$) < 0.0001, P($Mpz^{Cre/+}$:$Ralb^{fl/fl}$ vs. $Mpz^{Cre/+}$:$Rala^{fl/fl}$:$Ralb^{fl/fl}$) < 0.0001.

**F.** $n$ = 5 animals per genotype; one-way ANOVA, Tukey's multiple comparisons test; mean ± SEM: control = 4,009 ± 134.6, $Mpz^{Cre/+}$:$Rala^{fl/fl}$ = 4,212 ± 70.03, $Mpz^{Cre/+}$:$Ralb^{fl/fl}$ = 4,038 ± 69.99, $Mpz^{Cre/+}$:$Rala^{fl/fl}$:$Ralb^{fl/fl}$ = 2,202 ± 280.3; $F_{(3,\ 16)}$ = 33.65, P < 0.0001, P(control vs. $Mpz^{Cre/+}$:$Rala^{fl/fl}$) = 0.8144, P(control vs. $Mpz^{Cre/+}$:$Ralb^{fl/fl}$) = 0.9993, P(control vs. $Mpz^{Cre/+}$:$Rala^{fl/fl}$:$Ralb^{fl/fl}$) < 0.0001, P($Mpz^{Cre/+}$:$Rala^{fl/fl}$ vs. $Mpz^{Cre/+}$:$Ralb^{fl/fl}$) = 0.8725, P($Mpz^{Cre/+}$:$Rala^{fl/fl}$ vs. $Mpz^{Cre/+}$:$Rala^{fl/fl}$:$Ralb^{fl/fl}$) < 0.0001, P($Mpz^{Cre/+}$:$Ralb^{fl/fl}$ vs. $Mpz^{Cre/+}$:$Rala^{fl/fl}$:$Ralb^{fl/fl}$) < 0.0001.

**G.** $n$ = 5 animals per genotype; one-way ANOVA, Tukey's multiple comparisons test; mean ± SEM: control = 4,009 ± 134.5, $Mpz^{Cre/+}$:$Rala^{fl/fl}$ = 4,213 ± 70.05, $Mpz^{Cre/+}$:$Ralb^{fl/fl}$ = 4,038 ± 69.81, $Mpz^{Cre/+}$:$Rala^{fl/fl}$:$Ralb^{fl/fl}$ = 2,362 ± 290.2; $F_{(3,\ 16)}$ = 26.81, P < 0.0001, P(control vs. $Mpz^{Cre/+}$:$Rala^{fl/fl}$) = 0.8254, P(control vs. $Mpz^{Cre/+}$:$Ralb^{fl/fl}$) = 0.9993, P(control vs. $Mpz^{Cre/+}$:$Rala^{fl/fl}$:$Ralb^{fl/fl}$) < 0.0001, P($Mpz^{Cre/+}$:$Rala^{fl/fl}$ vs. $Mpz^{Cre/+}$:$Ralb^{fl/fl}$) = 0.8809, P($Mpz^{Cre/+}$:$Rala^{fl/fl}$ vs. $Mpz^{Cre/+}$:$Rala^{fl/fl}$:$Ralb^{fl/fl}$) < 0.0001, P($Mpz^{Cre/+}$:$Ralb^{fl/fl}$ vs. $Mpz^{Cre/+}$:$Rala^{fl/fl}$:$Ralb^{fl/fl}$) < 0.0001.

**H.** $n$ = 5 animals per genotype; one-way ANOVA, Tukey's multiple comparisons test; mean ± SEM: control = 0.2 ± 0.2, $Mpz^{Cre/+}$:$Rala^{fl/fl}$ = 0.2 ± 0.2, $Mpz^{Cre/+}$:$Ralb^{fl/fl}$ = 0.4 ± 0.245, $Mpz^{Cre/+}$:$Rala^{fl/fl}$:$Ralb^{fl/fl}$ = 160.6 ± 15.62; $F_{(3,\ 16)}$ = 105.4, P < 0.0001, P(control vs. $Mpz^{Cre/+}$:$Rala^{fl/fl}$) > 0.9999, P(control vs. $Mpz^{Cre/+}$:$Ralb^{fl/fl}$) > 0.9999, P(control vs. $Mpz^{Cre/+}$:$Rala^{fl/fl}$:$Ralb^{fl/fl}$) < 0.0001, P($Mpz^{Cre/+}$:$Rala^{fl/fl}$ vs. $Mpz^{Cre/+}$:$Ralb^{fl/fl}$) > 0.9999, P($Mpz^{Cre/+}$:$Rala^{fl/fl}$ vs. $Mpz^{Cre/+}$:$Rala^{fl/fl}$:$Ralb^{fl/fl}$) < 0.0001, P($Mpz^{Cre/+}$:$Ralb^{fl/fl}$ vs. $Mpz^{Cre/+}$:$Rala^{fl/fl}$:$Ralb^{fl/fl}$) < 0.0001.

**I.** $n$ = 5 animals per genotype; one-way ANOVA, Tukey's multiple comparisons test; mean ± SEM: control = 0.0 ± 0.0, $Mpz^{Cre/+}$:$Rala^{fl/fl}$ = 0.0 ± 0.0, $Mpz^{Cre/+}$:$Ralb^{fl/fl}$ = 0.0 ± 0.0, $Mpz^{Cre/+}$:$Rala^{fl/fl}$:$Ralb^{fl/fl}$ = 8.639 ± 2.288; $F_{(3,\ 16)}$ = 14.25, P < 0.0001, P(control vs. $Mpz^{Cre/+}$:$Rala^{fl/fl}$) > 0.9999, P(control vs. $Mpz^{Cre/+}$:$Ralb^{fl/fl}$) > 0.9999, P(control vs. $Mpz^{Cre/+}$:$Rala^{fl/fl}$:$Ralb^{fl/fl}$) = 0.0003, P($Mpz^{Cre/+}$:$Rala^{fl/fl}$ vs. $Mpz^{Cre/+}$:$Ralb^{fl/fl}$) > 0.9999, P($Mpz^{Cre/+}$:$Rala^{fl/fl}$ vs. $Mpz^{Cre/+}$:$Rala^{fl/fl}$:$Ralb^{fl/fl}$) = 0.0003, P($Mpz^{Cre/+}$:$Ralb^{fl/fl}$ vs. $Mpz^{Cre/+}$:$Rala^{fl/fl}$:$Ralb^{fl/fl}$) = 0.0003.

*Figure 5*

**B.** $n$ = 6 animals ($Mpz^{Cre/+}$:$Rala^{fl/fl}$, $Rala^{fl/fl}$:$Ralb^{-/-}$, $Mpz^{Cre/+}$:$Rala^{fl/fl}$:$Ralb^{-/-}$) or 7 animals (control) per genotype; one-way

ANOVA, Tukey's multiple comparisons test; mean ± SEM: control = 523.4 ± 21.79, $Mpz^{Cre/+}$:$Rala^{fl/fl}$ = 641.2 ± 11.56, $Rala^{fl/fl}$:$Ralb^{-/-}$ = 655.4 ± 32.16, $Mpz^{Cre/+}$:$Rala^{fl/fl}$:$Ralb^{-/-}$ = 633.6 ± 36.94; $F_{(3, 21)}$ = 5.402, P = 0.0065, P(control vs. $Mpz^{Cre/+}$:$Rala^{fl/fl}$) = 0.0241, P(control vs. $Rala^{fl/fl}$:$Ralb^{-/-}$) = 0.0103, P(control vs. $Mpz^{Cre/+}$:$Rala^{fl/fl}$:$Ralb^{-/-}$) = 0.0371, P($Mpz^{Cre/+}$:$Rala^{fl/fl}$ vs. $Rala^{fl/fl}$:$Ralb^{-/-}$) = 0.9828, P($Mpz^{Cre/+}$:$Rala^{fl/fl}$ vs. $Mpz^{Cre/+}$:$Rala^{fl/fl}$:$Ralb^{-/-}$) = 0.9973, P($Rala^{fl/fl}$:$Ralb^{-/-}$ vs. $Mpz^{Cre/+}$:$Rala^{fl/fl}$:$Ralb^{-/-}$) = 0.9429.

**C.** n = 6 animals ($Mpz^{Cre/+}$:$Rala^{fl/fl}$, $Rala^{fl/fl}$:$Ralb^{-/-}$, $Mpz^{Cre/+}$:$Rala^{fl/fl}$:$Ralb^{-/-}$) or 7 animals (control) per genotype; one-way ANOVA, Tukey's multiple comparisons test; mean ± SEM: control = 397 ± 14.52, $Mpz^{Cre/+}$:$Rala^{fl/fl}$ = 496.7 ± 42.4, $Rala^{fl/fl}$:$Ralb^{-/-}$ = 514.2 ± 11.01, $Mpz^{Cre/+}$:$Rala^{fl/fl}$:$Ralb^{-/-}$ = 486.5 ± 33.34; $F_{(3, 21)}$ = 3.852, P = 0.0243, P(control vs. $Mpz^{Cre/+}$:$Rala^{fl/fl}$) = 0.0733, P(control vs. $Rala^{fl/fl}$:$Ralb^{-/-}$) = 0.0284, P(control vs. $Mpz^{Cre/+}$:$Rala^{fl/fl}$:$Ralb^{-/-}$) = 0.1222, P($Mpz^{Cre/+}$:$Rala^{fl/fl}$ vs. $Rala^{fl/fl}$:$Ralb^{-/-}$) = 0.9709, P($Mpz^{Cre/+}$:$Rala^{fl/fl}$ vs. $Mpz^{Cre/+}$:$Rala^{fl/fl}$:$Ralb^{-/-}$) = 0.9940, P($Rala^{fl/fl}$:$Ralb^{-/-}$ vs. $Mpz^{Cre/+}$:$Rala^{fl/fl}$:$Ralb^{-/-}$) = 0.8978.

**D.** n = 6 animals ($Mpz^{Cre/+}$:$Rala^{fl/fl}$, $Rala^{fl/fl}$:$Ralb^{-/-}$, $Mpz^{Cre/+}$:$Rala^{fl/fl}$:$Ralb^{-/-}$) or 7 animals (control) per genotype; one-way ANOVA, Tukey's multiple comparisons test; mean ± SEM: control = 4.304 ± 0.566, $Mpz^{Cre/+}$:$Rala^{fl/fl}$ = 3.886 ± 0.477, $Rala^{fl/fl}$:$Ralb^{-/-}$ = 4.583 ± 0.509, $Mpz^{Cre/+}$:$Rala^{fl/fl}$:$Ralb^{-/-}$ = 9.47 ± 1.046; $F_{(3, 21)}$ = 14.41, P < 0.0001, P(control vs. $Mpz^{Cre/+}$:$Rala^{fl/fl}$) = 0.9705, P(control vs. $Rala^{fl/fl}$:$Ralb^{-/-}$) = 0.9909, P(control vs. $Mpz^{Cre/+}$:$Rala^{fl/fl}$:$Ralb^{-/-}$) = 0.0001, P($Mpz^{Cre/+}$:$Rala^{fl/fl}$ vs. $Rala^{fl/fl}$:$Ralb^{-/-}$) = 0.8925, P($Mpz^{Cre/+}$:$Rala^{fl/fl}$ vs. $Mpz^{Cre/+}$:$Rala^{fl/fl}$:$Ralb^{-/-}$) < 0.0001, P($Rala^{fl/fl}$:$Ralb^{-/-}$ vs. $Mpz^{Cre/+}$:$Rala^{fl/fl}$:$Ralb^{-/-}$) = 0.0003.

**F.** n = 7 animals (control, $Mpz^{Cre/+}$:$Rala^{fl/fl}$) or 6 animals ($Rala^{fl/fl}$:$Ralb^{-/-}$, $Mpz^{Cre/+}$:$Rala^{fl/fl}$:$Ralb^{-/-}$) per genotype; one-way ANOVA, Tukey's multiple comparisons test; mean ± SEM = control = 0.243 ± 0.096, $Mpz^{Cre/+}$:$Rala^{fl/fl}$ = 0.378 ± 0.145, $Rala^{fl/fl}$:$Ralb^{-/-}$ = 0.215 ± 0.122, $Mpz^{Cre/+}$:$Rala^{fl/fl}$:$Ralb^{-/-}$ = 0.276 ± 0.105; $F_{(3, 22)}$ = 0.3682, P = 0.7767, P(control vs. $Mpz^{Cre/+}$:$Rala^{fl/fl}$) = 0.8389, P(control vs. $Rala^{fl/fl}$:$Ralb^{-/-}$) = 0.9984, P(control vs. $Mpz^{Cre/+}$:$Rala^{fl/fl}$:$Ralb^{-/-}$) = 0.9971, P($Mpz^{Cre/+}$:$Rala^{fl/fl}$ vs. $Rala^{fl/fl}$:$Ralb^{-/-}$) = 0.7719, P($Mpz^{Cre/+}$:$Rala^{fl/fl}$ vs. $Mpz^{Cre/+}$:$Rala^{fl/fl}$:$Ralb^{-/-}$) = 0.9310, P($Rala^{fl/fl}$:$Ralb^{-/-}$ vs. $Mpz^{Cre/+}$:$Rala^{fl/fl}$:$Ralb^{-/-}$) = 0.9850.

### Figure 6
**B.** n = 5 animals per genotype; one-sample t test; mean ± SEM: control = 0.0 ± 0.0, $Mpz^{Cre/+}$:$Rala^{fl/fl}$ = 0.0 ± 0.0, $Rala^{fl/fl}$:$Ralb^{-/-}$ = 0.0 ± 0.0, $Mpz^{Cre/+}$:$Rala^{fl/fl}$:$Ralb^{-/-}$ = 102.4 ± 6.547; t = 15.64, degrees of freedom (df) = 4; P < 0.0001.

**D.** n = 4 samples, each consisting of sciatic nerves of three animals; unpaired two-tailed t test; mean ± SEM: control = 1 ± 0.1068, $Mpz^{Cre/+}$:$Rala^{fl/fl}$:$Ralb^{-/-}$ = 1.795 ± 0.0907; t = 5.675; df = 6; P = 0.0013.

**E.** n = 4 samples, each consisting of sciatic nerves of three animals; unpaired two-tailed t test; mean ± SEM: control = 1 ± 0.1565, $Mpz^{Cre/+}$:$Rala^{fl/fl}$:$Ralb^{-/-}$ = 0.663 ± 0.04693; t = 2.063; df = 6; P = 0.0847.

### Figure 7
**B.** n = 6 animals, ≥100 cells per animal, two independent experiments unpaired two-tailed t test; mean ± SEM: control =

2.342 ± 0.099, $Mpz^{Cre/+}$:$Rala^{fl/fl}$:$Ralb^{-/-}$ = 0.459 ± 0.130; t = 11.5; df = 10; P < 0.0001.

**C.** n = 6 animals, ≥100 cells per animal, two independent experiments; unpaired two-tailed t test; mean ± SEM: control = 1.555 ± 0.072, $Mpz^{Cre/+}$:$Rala^{fl/fl}$:$Ralb^{-/-}$ = 0.985 ± 0.122; t = 4.018; df = 10; P = 0.0024.

**D.** n = 5 animals, ≥100 cells per animal, two independent experiments; unpaired two-tailed t test; mean ± SEM: control = 73.6 ± 1.832, $Mpz^{Cre/+}$:$Rala^{fl/fl}$:$Ralb^{-/-}$ = 62.32 ± 2.151; t = 3.99; df = 8; P = 0.004.

### Figure 8
**A.** n = 6 animals, ≥100 cells per animal, two independent experiments with three animals each; multiple unpaired two-tailed t tests, Holm–Sidak method for multiple comparisons correction; mean ± SEM: control = 88.22 ± 7.766, $Mpz^{Cre/+}$:$Rala^{fl/fl}$:$Ralb^{-/-}$ = 85.76 ± 4.69; t = 0.2282; df = 62; P = 0.9677.

**B.** n = 6 animals, ≥100 cells per animal, two independent experiments with three animals each; multiple unpaired two-tailed t tests, Holm–Sidak method for multiple comparisons correction; mean ± SEM: control = 101.6 ± 9.305, $Mpz^{Cre/+}$:$Rala^{fl/fl}$:$Ralb^{-/-}$ = 40.26 ± 7.82; t = 5.703; df = 62; P < 0.0001.

**C.** n = 6 animals, ≥100 cells per animal, two independent experiments with three animals each; multiple unpaired two-tailed t tests, Holm–Sidak method for multiple comparisons correction; mean ± SEM: control = 104.1 ± 9.359, $Mpz^{Cre/+}$:$Rala^{fl/fl}$:$Ralb^{-/-}$ = 102.6 ± 10.61; t = 0.1397; df = 62; P = 0.9677.

**D.** n = 6 animals, ≥100 cells per animal, two independent experiments with three animals each; multiple unpaired two-tailed t tests, Holm–Sidak method for multiple comparisons correction; mean ± SEM: control = 100.8 ± 2.609, $Mpz^{Cre/+}$:$Rala^{fl/fl}$:$Ralb^{-/-}$ = 83.49 ± 6.586; t = 1.611; df = 62; P = 0.3005.

**E.** n = 7 animals, ≥100 cells per animal, two independent experiments with three or four animals each; multiple unpaired two-tailed t tests, Holm–Sidak method for multiple comparisons correction; mean ± SEM: control = 106.8 ± 7.568, $Mpz^{Cre/+}$:$Rala^{fl/fl}$:$Ralb^{-/-}$ = 68.14 ± 4.856; t = 3.881; df = 62; P = 0.001.

**F.** n = 6 animals, ≥100 cells per animal, two independent experiments with three animals each; multiple unpaired two-tailed t tests, Holm–Sidak method for multiple comparisons correction; mean ± SEM: control = 111.2 ± 6.548, $Mpz^{Cre/+}$:$Rala^{fl/fl}$:$Ralb^{-/-}$ = 44.69 ± 9.007; t = 6.179; df = 62; P < 0.0001.

### Figure S1
n = ≥30 animals per genotype.

### Figure S2
**A.** n = 5 animals, onion bulb–like structures were observed on each animal.

**B.** n = 5 animals (control, $Rala^{fl/fl}$:$Ralb^{-/-}$, $Mpz^{Cre/+}$:$Rala^{fl/fl}$:$Ralb^{-/-}$ at 2 mo and 1 yr; and $Mpz^{Cre/+}$:$Rala^{fl/fl}$ at 2 mo) or 4 animals (only $Mpz^{Cre/+}$:$Rala^{fl/fl}$ at 1 yr) per genotype, two-way ANOVA, Tukey's multiple comparisons test; mean ± SEM: $control_{2\ mo}$ = 2 ± 1.304, $Mpz^{Cre/+}$:$Rala^{fl/fl}_{2\ mo}$ = 4.8 ± 2.035, $Rala^{fl/fl}$:$Ralb^{-/-}_{2\ mo}$ = 10.2 ± 3.089, $Mpz^{Cre/+}$:$Rala^{fl/fl}$:$Ralb^{-/-}_{2\ mo}$ = 174.6 ± 20.02, $control_{1\ yr}$ = 23.4 ± 4.675, $Mpz^{Cre/+}$:$Rala^{fl/fl}_{1\ yr}$ = 28.5 ± 2.661, $Rala^{fl/fl}$:$Ralb^{-/-}_{1\ yr}$ = 24 ± 2.739, $Mpz^{Cre/+}$:$Rala^{fl/fl}$:

$Ralb^{-/-}_{1\text{ yr}}$ = 380 ± 56.36; $F_{(3,\,31)\text{genotype}}$ = 72.6, P(genotype) < 0.0001, $F_{(1,\,31)\text{age}}$ = 18.1, P(age) = 0.0002; P(control$_{2\text{ mo}}$ vs. $Mpz^{Cre/+}{:}Rala^{fl/fl}_{2\text{ mo}}$) > 0.9999, P(control$_{2\text{ mo}}$ vs. $Rala^{fl/fl}{:}Ralb^{-/-}_{2\text{ mo}}$) > 0.9999, P(control$_{2\text{ mo}}$ vs. $Mpz^{Cre/+}{:}Rala^{fl/fl}{:}Ralb^{-/-}_{2\text{ mo}}$) < 0.0001, P($Mpz^{Cre/+}{:}Rala^{fl/fl}_{2\text{ mo}}$ vs. $Rala^{fl/fl}{:}Ralb^{-/-}_{2\text{ mo}}$) > 0.9999, P($Mpz^{Cre/+}{:}Rala^{fl/fl}_{2\text{ mo}}$ vs. $Mpz^{Cre/+}{:}Rala^{fl/fl}{:}Ralb^{-/-}_{2\text{ mo}}$) = 0.0001, P($Rala^{fl/fl}{:}Ralb^{-/-}_{2\text{ mo}}$ vs. $Mpz^{Cre/+}{:}Rala^{fl/fl}{:}Ralb^{-/-}_{2\text{ mo}}$) = 0.0002; P(control$_{1\text{ yr}}$ vs. $Mpz^{Cre/+}{:}Rala^{fl/fl}_{1\text{ yr}}$) > 0.9999, P(control$_{1\text{ yr}}$ vs. $Rala^{fl/fl}{:}Ralb^{-/-}_{1\text{ yr}}$) > 0.9999, P(control$_{1\text{ yr}}$ vs. $Mpz^{Cre/+}{:}Rala^{fl/fl}{:}Ralb^{-/-}_{1\text{ yr}}$) < 0.0001, P($Mpz^{Cre/+}{:}Rala^{fl/fl}_{1\text{ yr}}$ vs. $Rala^{fl/fl}{:}Ralb^{-/-}_{1\text{ yr}}$) > 0.9999, P($Mpz^{Cre/+}{:}Rala^{fl/fl}_{1\text{ yr}}$ vs. $Mpz^{Cre/+}{:}Rala^{fl/fl}{:}Ralb^{-/-}_{1\text{ yr}}$) < 0.0001, P($Rala^{fl/fl}{:}Ralb^{-/-}_{1\text{ yr}}$ vs. $Mpz^{Cre/+}{:}Rala^{fl/fl}{:}Ralb^{-/-}_{1\text{ yr}}$) < 0.0001; P(control$_{2\text{ mo}}$ vs. control$_{1\text{ yr}}$) = 0.9964, P($Mpz^{Cre/+}{:}Rala^{fl/fl}_{2\text{ mo}}$ vs. $Mpz^{Cre/+}{:}Rala^{fl/fl}_{1\text{ yr}}$) = 0.9953, P($Rala^{fl/fl}{:}Ralb^{-/-}_{2\text{ mo}}$ vs. $Rala^{fl/fl}{:}Ralb^{-/-}_{1\text{ yr}}$) = 0.9998, P($Mpz^{Cre/+}{:}Rala^{fl/fl}{:}Ralb^{-/-}_{2\text{ mo}}$ vs. $Mpz^{Cre/+}{:}Rala^{fl/fl}{:}Ralb^{-/-}_{1\text{ yr}}$) < 0.0001.

**C.** $n$ = 5 animals per genotype, one cross section per animal. One-way ANOVA, Tukey's multiple comparisons test; mean ± SEM: control = 2.633 ± 0.247, $Mpz^{Cre/+}{:}Rala^{fl/fl}$ = 3.077 ± 0.185, $Rala^{fl/fl}{:}Ralb^{-/-}$ = 2.769 ± 0.28, $Mpz^{Cre/+}{:}Rala^{fl/fl}{:}Ralb^{-/-}$ = 5.771 ± 0.271; $F_{(3,\,16)}$ = 35.67, P < 0.0001, P(control vs. $Mpz^{Cre/+}{:}Rala^{fl/fl}$) = 0.5964, P(control vs. $Rala^{fl/fl}{:}Ralb^{-/-}$) = 0.9793, P(control vs. $Mpz^{Cre/+}{:}Rala^{fl/fl}{:}Ralb^{-/-}$) < 0.0001, P($Mpz^{Cre/+}{:}Rala^{fl/fl}$ vs. $Rala^{fl/fl}{:}Ralb^{-/-}$) = 0.8168, P($Mpz^{Cre/+}{:}Rala^{fl/fl}$ vs. $Mpz^{Cre/+}{:}Rala^{fl/fl}{:}Ralb^{-/-}$) < 0.0001, P($Rala^{fl/fl}{:}Ralb^{-/-}$ vs. $Mpz^{Cre/+}{:}Rala^{fl/fl}{:}Ralb^{-/-}$) < 0.0001.

**D.** $n$ = 4 ($Mpz^{Cre/+}{:}Rala^{fl/fl}$) or 5 (control, $Rala^{fl/fl}{:}Ralb^{-/-}$, $Mpz^{Cre/+}{:}Rala^{fl/fl}{:}Ralb^{-/-}$) animals per genotype, one cross section per animal. One-way ANOVA, Tukey's multiple comparisons test; mean ± SEM: control = 3.225 ± 0.611, $Mpz^{Cre/+}{:}Rala^{fl/fl}$ = 2.94 ± 0.418, $Rala^{fl/fl}{:}Ralb^{-/-}$ = 3.821 ± 0.951, $Mpz^{Cre/+}{:}Rala^{fl/fl}{:}Ralb^{-/-}$ = 6.961 ± 0.923; $F_{(3,\,15)}$ = 5.564, P = 0.009, P(control vs. $Mpz^{Cre/+}{:}Rala^{fl/fl}$) = 0.9945, P(control vs. $Rala^{fl/fl}{:}Ralb^{-/-}$) = 0.9463, P(control vs. $Mpz^{Cre/+}{:}Rala^{fl/fl}{:}Ralb^{-/-}$) = 0.0177, P($Mpz^{Cre/+}{:}Rala^{fl/fl}$ vs. $Rala^{fl/fl}{:}Ralb^{-/-}$) = 0.8708, P($Mpz^{Cre/+}{:}Rala^{fl/fl}$ vs. $Mpz^{Cre/+}{:}Rala^{fl/fl}{:}Ralb^{-/-}$) = 0.016, P($Rala^{fl/fl}{:}Ralb^{-/-}$ vs. $Mpz^{Cre/+}{:}Rala^{fl/fl}{:}Ralb^{-/-}$) = 0.0505.

## Figure S3

**A.** $n$ = 5 animals (control, $Rala^{fl/fl}{:}Ralb^{-/-}$, $Mpz^{Cre/+}{:}Rala^{fl/fl}{:}Ralb^{-/-}$) or 4 animals ($Mpz^{Cre/+}{:}Rala^{fl/fl}$) per genotype, ≥100 axons per animal selected from four random fields, one-way ANOVA, Tukey's multiple comparisons test; mean ± SEM: control = 0.6566 ± 0.003, $Mpz^{Cre/+}{:}Rala^{fl/fl}$ = 0.6545 ± 0.007, $Rala^{fl/fl}{:}Ralb^{-/-}$ = 0.6132 ± 0.007, $Mpz^{Cre/+}{:}Rala^{fl/fl}{:}Ralb^{-/-}$ = 0.6997 ± 0.016; $F_{(3,\,15)}$ = 13.91, P = 0.0001, P(control vs. $Mpz^{Cre/+}{:}Rala^{fl/fl}$) = 0.9986, P(control vs. $Rala^{fl/fl}{:}Ralb^{-/-}$) = 0.0251, P(control vs. $Mpz^{Cre/+}{:}Rala^{fl/fl}{:}Ralb^{-/-}$) = 0.0265, P($Mpz^{Cre/+}{:}Rala^{fl/fl}$ vs. $Rala^{fl/fl}{:}Ralb^{-/-}$) = 0.0480, P($Mpz^{Cre/+}{:}Rala^{fl/fl}$ vs. $Mpz^{Cre/+}{:}Rala^{fl/fl}{:}Ralb^{-/-}$) = 0.0281, P($Rala^{fl/fl}{:}Ralb^{-/-}$ vs. $Mpz^{Cre/+}{:}Rala^{fl/fl}{:}Ralb^{-/-}$) < 0.0001.

**B.** $n$ = 5 animals (control, $Rala^{fl/fl}{:}Ralb^{-/-}$, $Mpz^{Cre/+}{:}Rala^{fl/fl}{:}Ralb^{-/-}$) or 4 animals ($Mpz^{Cre/+}{:}Rala^{fl/fl}$) per genotype, ≥100 axons per animal selected from four random fields, two-way ANOVA, Dunnett's multiple comparisons test; mean ± SEM: control$_{0-1\text{ μm}}$ = 0.0 ± 0.0, control$_{1-2\text{ μm}}$ = 4.781 ± 0.9088, control$_{2-3\text{ μm}}$ = 21.39 ± 0.9408, control$_{3-4\text{ μm}}$ = 21.98 ± 3.249, control$_{4-5\text{ μm}}$ = 17.87 ± 1.665, control$_{5-6\text{ μm}}$ = 11.56 ± 2.137, control$_{6-7\text{ μm}}$ = 10.5 ± 2.258, control$_{>7\text{ μm}}$ = 11.92 ± 1.078, $Mpz^{Cre/+}{:}Rala^{fl/fl}_{0-1\text{ μm}}$ = 0.0 ± 0.0, $Mpz^{Cre/+}{:}Rala^{fl/fl}_{1-2\text{ μm}}$ = 4.796 ± 1.101, $Mpz^{Cre/+}{:}Rala^{fl/fl}_{2-3\text{ μm}}$ = 20.2 ± 2.024, $Mpz^{Cre/+}{:}Rala^{fl/fl}_{3-4\text{ μm}}$ = 22.35 ± 4.245, $Mpz^{Cre/+}{:}Rala^{fl/fl}_{4-5\text{ μm}}$ = 20.44 ± 2.144, $Mpz^{Cre/+}{:}Rala^{fl/fl}_{5-6\text{ μm}}$ = 15.84 ± 2.563, $Mpz^{Cre/+}{:}Rala^{fl/fl}_{6-7\text{ μm}}$ = 5.931 ± 1.124, $Mpz^{Cre/+}{:}Rala^{fl/fl}_{>7\text{ μm}}$ = 10.44 ± 2.338, $Rala^{fl/fl}{:}Ralb^{-/-}_{0-1\text{ μm}}$ = 0.0 ± 0.0, $Rala^{fl/fl}{:}Ralb^{-/-}_{1-2\text{ μm}}$ = 9.242 ± 2.269, $Rala^{fl/fl}{:}Ralb^{-/-}_{2-3\text{ μm}}$ = 20.78 ± 2.049, $Rala^{fl/fl}{:}Ralb^{-/-}_{3-4\text{ μm}}$ = 22.5 ± 3.688, $Rala^{fl/fl}{:}Ralb^{-/-}_{4-5\text{ μm}}$ = 17.35 ± 0.7545, $Rala^{fl/fl}{:}Ralb^{-/-}_{5-6\text{ μm}}$ = 10.9 ± 2.324, $Rala^{fl/fl}{:}Ralb^{-/-}_{6-7\text{ μm}}$ = 6.818 ± 0.8219, $Rala^{fl/fl}{:}Ralb^{-/-}_{>7\text{ μm}}$ = 12.4 ± 2.956, $Mpz^{Cre/+}{:}Rala^{fl/fl}{:}Ralb^{-/-}_{0-1\text{ μm}}$ = 0.73 ± 0.1842, $Mpz^{Cre/+}{:}Rala^{fl/fl}{:}Ralb^{-/-}_{1-2\text{ μm}}$ = 15.67 ± 1.948, $Mpz^{Cre/+}{:}Rala^{fl/fl}{:}Ralb^{-/-}_{2-3\text{ μm}}$ = 27.11 ± 3.75, $Mpz^{Cre/+}{:}Rala^{fl/fl}{:}Ralb^{-/-}_{3-4\text{ μm}}$ = 30.88 ± 3.417, $Mpz^{Cre/+}{:}Rala^{fl/fl}{:}Ralb^{-/-}_{4-5\text{ μm}}$ = 16.2 ± 0.7165, $Mpz^{Cre/+}{:}Rala^{fl/fl}{:}Ralb^{-/-}_{5-6\text{ μm}}$ = 5.004 ± 1.615, $Mpz^{Cre/+}{:}Rala^{fl/fl}{:}Ralb^{-/-}_{6-7\text{ μm}}$ = 3.198 ± 0.9082, $Mpz^{Cre/+}{:}Rala^{fl/fl}{:}Ralb^{-/-}_{>7\text{ μm}}$ = 1.211 ± 0.7536; $F_{(21,\,120)}$ interaction genotype × axonal size = 3.87, $P_{\text{interaction}}$ < 0.0001, P(control vs. $Mpz^{Cre/+}{:}Rala^{fl/fl}$)$_{0-1\text{ μm}}$ = 0.9999, P(control vs. $Rala^{fl/fl}{:}Ralb^{-/-}$)$_{0-1\text{ μm}}$ = 0.9999, P(control vs. $Mpz^{Cre/+}{:}Rala^{fl/fl}{:}Ralb^{-/-}$)$_{0-1\text{ μm}}$ = 0.9886, P(control vs. $Mpz^{Cre/+}{:}Rala^{fl/fl}$)$_{1-2\text{ μm}}$ = 0.9999, P(control vs. $Rala^{fl/fl}{:}Ralb^{-/-}$)$_{1-2\text{ μm}}$ = 0.2813, P(control vs. $Mpz^{Cre/+}{:}Rala^{fl/fl}{:}Ralb^{-/-}$)$_{1-2\text{ μm}}$ = 0.0007, P(control vs. $Mpz^{Cre/+}{:}Rala^{fl/fl}$)$_{2-3\text{ μm}}$ = 0.9621, P(control vs. $Rala^{fl/fl}{:}Ralb^{-/-}$)$_{2-3\text{ μm}}$ = 0.9935, P(control vs. $Mpz^{Cre/+}{:}Rala^{fl/fl}{:}Ralb^{-/-}$)$_{2-3\text{ μm}}$ = 0.1198, P(control vs. $Mpz^{Cre/+}{:}Rala^{fl/fl}$)$_{3-4\text{ μm}}$ = 0.9986, P(control vs. $Rala^{fl/fl}{:}Ralb^{-/-}$)$_{3-4\text{ μm}}$ = 0.9957, P(control vs. $Mpz^{Cre/+}{:}Rala^{fl/fl}{:}Ralb^{-/-}$)$_{3-4\text{ μm}}$ = 0.0067, P(control vs. $Mpz^{Cre/+}{:}Rala^{fl/fl}$)$_{4-5\text{ μm}}$ = 0.7323, P(control vs. $Rala^{fl/fl}{:}Ralb^{-/-}$)$_{4-5\text{ μm}}$ = 0.9959, P(control vs. $Mpz^{Cre/+}{:}Rala^{fl/fl}{:}Ralb^{-/-}$)$_{4-5\text{ μm}}$ = 0.8892, P(control vs. $Mpz^{Cre/+}{:}Rala^{fl/fl}$)$_{5-6\text{ μm}}$ = 0.3598, P(control vs. $Rala^{fl/fl}{:}Ralb^{-/-}$)$_{5-6\text{ μm}}$ = 0.9916, P(control vs. $Mpz^{Cre/+}{:}Rala^{fl/fl}{:}Ralb^{-/-}$)$_{5-6\text{ μm}}$ = 0.0621, P(control vs. $Mpz^{Cre/+}{:}Rala^{fl/fl}$)$_{6-7\text{ μm}}$ = 0.3073, P(control vs. $Rala^{fl/fl}{:}Ralb^{-/-}$)$_{6-7\text{ μm}}$ = 0.4324, P(control vs. $Mpz^{Cre/+}{:}Rala^{fl/fl}{:}Ralb^{-/-}$)$_{6-7\text{ μm}}$ = 0.0323, P(control vs. $Mpz^{Cre/+}{:}Rala^{fl/fl}$)$_{>7\text{ μm}}$ = 0.9294, P(control vs. $Rala^{fl/fl}{:}Ralb^{-/-}$)$_{>7\text{ μm}}$ = 0.9968, P(control vs. $Mpz^{Cre/+}{:}Rala^{fl/fl}{:}Ralb^{-/-}$)$_{>7\text{ μm}}$ = 0.0008.

## Online supplemental material

Fig. S1 shows the abnormal appearance of sciatic nerves of 2-mo-old RalA/B double-mutant mice compared with controls. Fig. S2 shows evidence of myelin aberrations and demyelination in adult RalA/B double-mutant mice. Fig. S3 shows the g-ratios and axonal size frequencies of 1-yr-old mutant and control mice.

## Acknowledgments

We thank the members of the Suter laboratory for data discussions and the ScopeM imaging facility of the ETH Zurich, Drs. Ned Mantei, and Cristina Fimiani for excellent technical support. We thank also Drs. Maria Laura Feltri and Lawrence Wrabetz for *Mpz*-Cre mice, Dr. Pascal Peschard and the late Dr.

Chris Marshall for Ral transgenic mice, and Dr. John Collard for the GST-p21–activated kinase-crib domain construct.

This work was supported by the Schweizerischer Nationalfonds zur Förderung der Wissenschaftlichen Forschung (to U. Suter).

The authors declare no competing financial interests.

Author contributions: A. Ommer designed and performed experiments, analyzed data, and wrote the manuscript. G. Figlia designed experiments, analyzed data and edited the manuscript. J.A. Pereira performed experiments and analyzed and reviewed all data. A.L. Datwyler, J. Gerber, and J. DeGeer performed experiments. G. Lalli provided crucial models and reagents. U. Suter supervised the work, designed experiments, and edited the manuscript. All authors commented on and approved the manuscript.

Submitted: 28 November 2018

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
