## [Reviewer comments · The Journal of Cell Biology]

Ral GTPases in Schwann Cells Promote Radial Axonal Sorting in the Peripheral Nervous System

Andrea Ommer, Gianluca Figlia, Jorge Pereira, Anna Lena Datwyler, Joanne Gerber, Jonathan DeGeer, Giovanna Lalli, and Ueli Suter

Corresponding Author(s): Ueli Suter, ETH Zürich

Review Timeline:

Submission Date:	2018-11-28
Editorial Decision:	2019-01-10
Revision Received:	2019-04-03
Editorial Decision:	2019-05-01
Revision Received:	2019-05-07
Accepted:	2019-05-15

Monitoring Editor: Marc Freeman

Scientific Editor: Tim Spencer

Transaction Report:

DOI: <https://doi.org/10.1083/jcb.201811150>

January 4, 2019

Re: JCB manuscript #201811150

Prof. Ueli Suter
ETH Zürich
Biology
Institute of Molecular Health Sciences
Otto-Stern-Weg 7
Zürich CH-8093
Switzerland

Dear Ueli,

Thank you for submitting your manuscript entitled "Ral GTPases Promote Radial Axonal Sorting via Exocyst-Dependent Control of Schwann Cell Protrusions". The manuscript was assessed by expert reviewers, whose comments are appended to this letter. We invite you to submit a revision if you can address the reviewers' key concerns, as outlined here.

You will see that while reviewer #2 supports publication of the paper essentially as-is, the other two reviewers, while enthusiastic about the work, have raised a number of concerns which will need to be addressed before the paper would be suitable for publication. In particular, rev#1 feels that further mechanistic insight is necessary and has suggested that one potential avenue of exploration would be to examine whether an interaction between the exocyst complex and Dlg1/Mtmr2 may be playing a role. We agree that a mechanistic extension such as this would be important for publication. This reviewer has also suggested that you examine the effects of RalA/B deletion at later stages. While we concur that this would be an interesting avenue of exploration, we do not feel that these experiments are strictly necessary for the current study so, while we would encourage the addition of such data, it will not be a requirement for resubmission. We hope that you will be able to provide a mechanistic extension and also address each of the other reviewer comments.

GENERAL GUIDELINES:

Text limits: Character count for an Article is < 40,000, not including spaces. Count includes title page, abstract, introduction, results, discussion, acknowledgments, and figure legends. Count does not include materials and methods, references, tables, or supplemental legends.

Figures: Articles may have up to 10 main text figures. Figures must be prepared according to the policies outlined in our Instructions to Authors, under Data Presentation, <http://jcb.rupress.org/site/misc/ifora.xhtml>. All figures in accepted manuscripts will be screened prior to publication.

IMPORTANT: It is JCB policy that if requested, original data images must be made available. Failure to provide original images upon request will result in unavoidable delays in publication. Please ensure that you have access to all original microscopy and blot data images before submitting your revision.

Supplemental information: There are strict limits on the allowable amount of supplemental data. Articles may have up to 5 supplemental figures. Up to 10 supplemental videos or flash animations are allowed. A summary of all supplemental material should appear at the end of the Materials and methods section.

The typical timeframe for revisions is three months; if submitted within this timeframe, novelty will not be reassessed at the final decision. Please note that papers are generally considered through only one revision cycle, so any revised manuscript will likely be either accepted or rejected.

Thank you for this interesting contribution to Journal of Cell Biology. You can contact us at the journal office with any questions, cellbio@rockefeller.edu or call (212) 327-8588.

Sincerely,

Marc Freeman, PhD
Monitoring Editor
JCB

Tim Spencer, PhD
Deputy Editor
Journal of Cell Biology
ORCID: 0000-0003-0716-9936

Reviewer #1 (Comments to the Authors (Required)):

In this paper the authors report the characterization of mouse mutants with Schwann cell specific ablation of RalA and B GTPases. Single RalA or B knock outs show normal nerve development, whereas double RalA and B knock outs specifically in Schwann cells display axonal sorting defects. The authors propose that Ral GTPases act through the exocyst complex to regulate Schwann cell protrusion, which is a key process in radial sorting.

The proposed role of RalA and B GTPase in regulating Schwann cell process extension in radial sorting represents a novel finding. However, this is a descriptive characterization of a radial sorting phenotype, which has been well documented over the last years. How RalA and B act through the exocyst complex and the interplay between Ral GTPases and Rac1, which also regulates Schwann cell process extension in Schwann cell development and radial sorting, remain to be assessed.

More specifically:

1- Figure 4 is redundant, as shows the same results as Figure 2, using P0-Cre transgene for both loci, RalA and B. It is a confirmatory finding that can be moved to Supplementary information.

2- Instead, starting from 2 months, abnormal myelination is observed, a feature that is not typical of axonal sorting defects. Is this a consequence of loss of additional RalA/B functions in nerve development? As the exocyst complex has been reported to interact with Dlg1 and Mtmr2, whose loss causes myelin abnormalities, this aspect should be further investigated. For instance, what happens if RalA/B ablation occurs later in postnatal nerve development?

3- Figure 7.

The transduction efficiency by Western blot and the titre or MOI used should be shown/indicated if different constructs with different effects are used and compared.

The authors suggest that RalA promotes Schwann cell process extension and elongation through the exocyst complex and not via RalBP1 and PLD1 effectors. This finding should be further investigated *in vivo*. Is the complex localization displaced/decreased in mutant nerves? Is the RalA/exocyst (Exoc2/c8) interaction occurring in Schwann cells in the nerve? Etc

Reviewer #2 (Comments to the Authors (Required)):

This study by Ommer and colleagues examine the role of the small GTPases RalA and RalB in Schwann cells of the developing and adult peripheral nervous system. RalA and RalB have been implicated in a range of cellular functions such as proliferation, vesicle targeting and receptor-mediated endocytosis. The proteins signal mainly downstream of AKT and Ras and target one or more of several known effectors. These include RBP1, PhospholipaseD1 and the exocyst components Exoc2 and Exoc8.

The authors demonstrate that Schwann cells lacking both RalA and RalB (Schwann cell-precursor specific ablation of RalA on a constitutive RalB^{-/-} background or Schwann cell-precursor specific ablation of both RalA and RalB) fail to properly segregate axons (impaired radial sorting) and persistence of non-myelinated sorted axons with a diameter over 1 micron. These developmental defects are not resolved with time and in fact result in complex defects in adult animals with evidence of demyelination/remyelination and axonal loss. The lack of RalA/B in Schwann cells does not affect their proliferation or survival.

The authors then provide convincing evidence that the observed defects are caused by a reduced ability of RalA/B^{-/-} Schwann cells to produce or stabilise radial processes. This inability of RalA/B^{-/-} Schwann cells to produce radial processes in culture could be rescued by constitutive active RalA and by constitutive active mutant RalA that had lost its ability to interact with PDL1 or RBP1. However, a constitutive RalA that could not interact with Exoc2 or Exoc8 did not rescue radial process formation in RalA/B mutant Schwann cells. These data strongly support a role for RalA/B signalling through the exocyst complex in radial process formation/stabilisation in normal Schwann cell development.

This is a well-executed study with carefully presented data that strongly support the main conclusions of this paper. I have no issues to raise.

Reviewer #3 (Comments to the Authors (Required)):

The paper by Ommer et al. describes convincingly a role for Ral GTPases in peripheral nerve development and provides a possible molecular mechanism for their role. The data are clear, solid and show that RalA and RalB are dispensable by themselves for nerve development, but their combined deletion in Schwann cells causes an arrest in radial sorting of axons. Morphological analysis suggests that the absence of Rals impedes the formation of Schwann cell protrusions, and data in isolated mutant Schwann cells in vitro supports this view by showing that the formation of lamellipodia is impaired. Using mutants lacking Ral ability to bind different effectors, the authors go on to show that binding to exocyst effectors -2 and -8 are required to rescue the in vitro Schwann cell phenotype. Overall, the paper is well done, well written and provides novel and important information on the molecular mechanisms that drives radial axonal sorting by Schwann cells during peripheral nerve development. The points described below would make the picture more complete and the role of the exocyst more convincing.

Major points:

- It would be more convincing if the rescue experiment is done in a system closer to the in vivo situation, i.e. a coculture DRG-SC system.
- It would be important to determine if the activation of Rac1 is altered in mutant Schwann cells

Minor point:

- Given the severity of the peripheral neuropathy, it is a bit surprising that the mice appear phenotypically normal, the authors may wish to comment on this.

"Answer to Reviewer Comments", Changes to the original manuscript are highlighted in the revised version

Reviewer #1 (Comments to the Authors (Required)):

1. Figure 4 is redundant, as shows the same results as Figure 2, using P0-Cre transgene for both loci, RalA and B. It is a confirmatory finding that can be moved to Supplementary information.

While we agree that Figure 4 is largely confirming the findings described in Figure 2, we feel that it adds substantially to the manuscript by showing that the observed defects are caused by loss of both Ral GTPases in Schwann cells specifically, independent of a potential function of RalB in other cell types. Therefore, we would like to keep Figure 4 as a full text figure, but we are willing to change it to a supplementary figure if the reviewer feels very strongly about this point.

2. Instead, starting from 2 months, abnormal myelination is observed, a feature that is not typical of axonal sorting defects. Is this a consequence of loss of additional RalA/B functions in nerve development? As the exocyst complex has been reported to interact with Dlg1 and Mtmr2, whose loss causes myelin abnormalities, this aspect should be further investigated. For instance, what happens if RalA/B ablation occurs later in postnatal nerve development?

We agree that the abnormal myelination observed in adult RalA/B double-mutant mice is an interesting feature that warrants further investigation. The mouse model employed in our current study does not permit, however, a thorough investigation of this potential late-onset phenotype due to the rather strong and persistent developmental defects. Detailed studies on this topic require experimental mice that allow inducible elimination of Ral GTPases. Such investigations are beyond the scope of our current study. Furthermore, we agree that a potential involvement of Dlg1 and Mtmr2 in the observed phenotype is conceivable. We have not been able to analyze this aspect in detail, mainly due to a lack of availability of suitable and reliable tools (see also above). Yet, we have added an appropriate comment in the Discussion section of the manuscript to emphasize potential connections.

3. Figure 7.
The transduction efficiency by Western blot and the titre or MOI used should be shown/indicated if different constructs with different effects are used and compared.

We have added the MOIs in the Material and Methods section. Western blot analyses were not feasible due to the limited number of cells that could be obtained from our preparations and the rather low infection rates. Nevertheless, since our analysis was performed considering infected- and expressing cells only, as assessed by immunostaining against the myc-tag carried by the corresponding proteins, we are confident that this limitation does not affect the conclusions drawn from the experiments.

4. The authors suggest that RalA promotes Schwann cell process extension and elongation through the exocyst complex and not via RalBP1 and PLD1 effectors. This finding should be further investigated in vivo. Is the complex localization displaced/decreased in mutant nerves? Is the RalA/exocyst (Exoc2/c8) interaction occurring in Schwann cells in the nerve? Etc

These are interesting and relevant questions that we tried to address as follows:

- a. Is the complex localization displaced in mutant nerves?

To tackle this question, we performed immunostaining on longitudinal sections of sciatic nerves of P5 control and double-mutant mice. Analyzing the sections with confocal microscopy, we did not observe striking differences in the localization of Exoc2 or Exoc8. However, we cannot exclude subtle differences due to the lack of highly specific reliable markers for the compartments to which Exoc2 and Exoc8 localize in vivo and the quantitative limitations of our analysis.

- b. Is the complex decreased in mutant nerves?
We analyzed the expression levels of Exoc2, Exoc8, and Exoc4 in P5 sciatic nerve lysates by Western blot. Unfortunately, the quality / specificity of the available reagents was such that we are not confident to draw firm conclusions from the data obtained.
- c. Is the RalA/exocyst (Exoc2/Exoc8) interaction occurring in Schwann cells in the nerve? Etc
Addressing this question precisely is experimentally very challenging. Such an in vivo analysis would have to include extensive and technically demanding experiments utilizing, for example, genetically-expressed Förster resonance energy transfer constructs for Ral GTPases and the exocyst complex (including extensive controls). This approach is beyond the scope of this study. In a less specialized setting, we used the GST-fused Ral-binding domain of RalBP1 to capture active Ral GTPases from sciatic nerve lysates to identify binding partners by Western Blot. These experiments were not successful with regard to the exocyst complex, since the complex precipitated in the negative-control condition (i.e. glutathione-S-transferase bound to glutathione-conjugated beads). Unfortunately, we were not able to overcome this technical problem.

Reviewer #2 (Comments to the Authors (Required)):

[...] I have no issues to raise.

Reviewer #3 (Comments to the Authors (Required)):

Major points:

1. It would be more convincing if the rescue experiment is done in a system closer to the in vivo situation, i.e. a coculture DRG-SC system.

We found no detectably deficient myelination in DRG explant cultures derived from $Mpz^{Cre+};Rala^{f/f};Ralb^{f/f}$ mice, rendering rescue experiments not possible in the available setting.

2. It would be important to determine if the activation of Rac1 is altered in mutant Schwann cells

We agree with this statement and have now successfully performed activity assays for Rac1 in sciatic nerve lysates. As this is a very important piece of information, we have added the results of these experiments to Figure 6 of our manuscript (note that in the process, we decided to split the original Figure 6 in two parts, the second part of which is now Figure 7). The Results section has also been extended accordingly, together with adjusting the Discussion.

Minor point:

Given the severity of the peripheral neuropathy, it is a bit surprising that the mice appear phenotypically normal, the authors may wish to comment on this.

We had already included all available data on this issue in the manuscript. We feel that additional interpretations would be too speculative and thus, we refrained from doing so.

May 1, 2019

RE: JCB Manuscript #201811150R

Prof. Ueli Suter
ETH Zürich
Biology
Institute of Molecular Health Sciences
Otto-Stern-Weg 7
Zürich CH-8093
Switzerland

Dear Ueli:

Thank you for submitting your revised manuscript entitled "Ral GTPases Promote Radial Axonal Sorting via Exocyst-Dependent Control of Schwann Cell Protrusions". The paper has now been assessed by the original reviewers #1 and #3. As you will see, reviewer #3 is now satisfied by your revisions but reviewer #1 still has some important concerns. This reviewer continues to feel that the support for the contention that RalA/B act via the exocyst complex to regulate developmental radial sorting is limited and s/he also contends that this is exacerbated by the discrepancy between the apparent/proposed function of RalBP1 and Rac1 in vivo and the fact that RalBP1 binding is dispensable for the culture assays, thereby preventing a clear interpretation of the molecular mechanisms which regulate RalA/B-mediated radial sorting.

With regard to the first issue, while we agree with the reviewer that your paper does not present clear evidence that exocyst complex function is necessary for myelination, your study does provide convincing evidence to link the RalA-exocyst connection with Schwann cell process extension. This data, combined with the fact that the accompanying paper shows that this connection is necessary for iPSC-derived sensory neuron myelination in culture, lead us to feel that this point is sufficiently demonstrated at this point.

However, we do agree with this reviewer that the proposed functions of RalBP1 and Rac1 are quite unclear, particularly given the new data showing that Rac1 activity (but not expression) is elevated in the mutant animals. While you have attempted to provide some context for this conundrum in the Discussion, we feel that this text still does not sufficiently clarify the mechanisms at work in this pathway. While we will not require further experiments to address this issue, we would like for you to provide a more thorough explanation for this discrepancy in the in vivo and culture scenarios and to clarify in detail what conclusions can and cannot be drawn from your data.

If these conditions are met, we would be happy to publish your paper in JCB pending final revisions necessary to meet our formatting guidelines (see details below). But please note that final acceptance will be contingent upon your responses to this final point.

A. MANUSCRIPT ORGANIZATION AND FORMATTING:

Full guidelines are available on our Instructions for Authors page, <http://jcb.rupress.org/submission-guidelines#revised>. **Submission of a paper that does not conform to JCB guidelines will delay the acceptance of your manuscript.**

1) Text limits: Character count for Articles and Tools is < 40,000, not including spaces. Count includes title page, abstract, introduction, results, discussion, acknowledgments, and figure legends. Count does not include materials and methods, references, tables, or supplemental legends.

2) Figures limits: Articles and Tools may have up to 10 main text figures.

3) Figure formatting: Scale bars must be present on all microscopy images, including inset magnifications. Molecular weight or nucleic acid size markers must be included on all gel electrophoresis.

4) Statistical analysis: Error bars on graphic representations of numerical data must be clearly described in the figure legend. The number of independent data points (n) represented in a graph must be indicated in the legend. Statistical methods should be explained in full in the materials and methods. For figures presenting pooled data the statistical measure should be defined in the figure legends. Please also be sure to indicate the statistical tests used in each of your experiments (both in the figure legend itself and in a separate methods section) as well as the parameters of the test (for example, if you ran a t-test, please indicate if it was one- or two-sided, etc.).

5) Materials and methods: Should be comprehensive and not simply reference a previous publication for details on how an experiment was performed. Please provide full descriptions (at least in brief) in the text for readers who may not have access to referenced manuscripts.

6) Please be sure to provide the sequences for all of your primers/oligos and RNAi constructs in the materials and methods. You must also indicate in the methods the source, species, and catalog numbers (where appropriate) for all of your antibodies.

7) Microscope image acquisition: The following information must be provided about the acquisition and processing of images:

- a. Make and model of microscope
- b. Type, magnification, and numerical aperture of the objective lenses
- c. Temperature
- d. imaging medium
- e. Fluorochromes
- f. Camera make and model
- g. Acquisition software
- h. Any software used for image processing subsequent to data acquisition. Please include details and types of operations involved (e.g., type of deconvolution, 3D reconstitutions, surface or volume rendering, gamma adjustments, etc.).

8) References: There is no limit to the number of references cited in a manuscript. References should be cited parenthetically in the text by author and year of publication. Abbreviate the names of journals according to PubMed.

9) Supplemental materials: There are strict limits on the allowable amount of supplemental data. Articles/Tools may have up to 5 supplemental figures. Please also note that tables, like figures, should be provided as individual, editable files. A summary of all supplemental material should

appear at the end of the Materials and methods section.

10) Conflict of interest statement: JCB requires inclusion of a statement in the acknowledgements regarding competing financial interests. If no competing financial interests exist, please include the following statement: "The authors declare no competing financial interests." If competing interests are declared, please follow your statement of these competing interests with the following statement: "The authors declare no further competing financial interests."

11) ORCID IDs: ORCID IDs are unique identifiers allowing researchers to create a record of their various scholarly contributions in a single place. At resubmission of your final files, please consider providing an ORCID ID for as many contributing authors as possible.

B. FINAL FILES:

-- High-resolution figure and video files: See our detailed guidelines for preparing your production-ready images, <http://jcb.rupress.org/fig-vid-guidelines>.

Thank you for this interesting contribution, we look forward to publishing your paper in Journal of Cell Biology.

Sincerely,

Marc Freeman, PhD
Monitoring Editor
JCB

Tim Spencer, PhD
Deputy Editor
Journal of Cell Biology

Reviewer #1 (Comments to the Authors (Required)):

The conclusion of this paper is quite strong as states that RalA/B act through exocyst complex to promote radial sorting. The exocyst involvement must be further supported somehow, if not in vivo at least ex vivo in Schwann cells and at the biochemical level.

This is even more important now in the revised version of the paper in which the increased Rac1-GTP observed in RalA/B KO nerves in vivo is not in line with the ex vivo experiment, which excludes RalBP1 involvement.

Reviewer #3 (Comments to the Authors (Required)):

The authors responded to my concerns.